# Bayesian Neural Controlled Differential Equations for Treatment Effect Estimation

**Konstantin Hess, Valentyn Melnychuk, Dennis Frauen & Stefan Feuerriegel**
Munich Center for Machine Learning
LMU Munich
`{k.hess,melnychuk,frauen,feuerriegel}@lmu.de`

## Abstract

Treatment effect estimation in continuous time is crucial for personalized medicine. However, existing methods for this task are limited to point estimates of the potential outcomes, whereas uncertainty estimates have been ignored. Needless to say, uncertainty quantification is crucial for reliable decision-making in medical applications. To fill this gap, we propose a novel *Bayesian neural controlled differential equation* (BNCDE) for treatment effect estimation in continuous time. In our BNCDE, the time dimension is modeled through a coupled system of neural controlled differential equations and neural stochastic differential equations, where the neural stochastic differential equations allow for tractable variational Bayesian inference. Thereby, for an assigned sequence of treatments, our BNCDE provides meaningful posterior predictive distributions of the potential outcomes. To the best of our knowledge, ours is the first tailored neural method to provide uncertainty estimates of treatment effects in continuous time. As such, our method is of direct practical value for promoting reliable decision-making in medicine.

## 1 Introduction

Personalized medicine seeks to choose treatments that improve a patient's future health trajectory. To this end, reliable estimates of treatment effects over time are needed (Allam et al., 2021; Bica et al., 2021). For example, in cancer therapy, a physician may base the decisions of applying chemotherapy on whether or not the expected health trajectory will improve after treatment.

In medicine, there is a growing interest in estimating treatment effects from patient trajectories using observational data (e.g., electronic health records) (Allam et al., 2021; Bica et al., 2021; Feuerriegel et al., 2024). Methods for this task should fulfill two requirements: (1) Existing methods typically require a patient's health trajectory to be recorded in regular time steps (e.g., Bica et al., 2020; Melnychuk et al., 2022). However, medical practice is highly volatile and dynamic, as patients may need immediate treatment. Hence, methods are needed that model patient trajectories not in discrete time (e.g., fixed daily or hourly time steps) but in **continuous time** (i.e., actual timestamps). (2) To ensure reliable decision-making, medicine is not only interested in point estimates but also the corresponding uncertainty (e.g., credible intervals) (Zampieri et al., 2021; Banerji et al., 2023). For example, rather than saying that a treatment is expected to reduce the size of a tumor by $x$, one is interested in whether the size of a tumor will reduce by $x$ with *95% probability, given the evidence of available data*. Hence, methods for treatment effect estimation must allow for **uncertainty quantification**. To the best of our knowledge, a tailored method that accounts for both (1) and (2) is still missing.

Several neural methods have been developed for individualized treatment effect estimation from observational data over time (see Section 2 for an overview). Here, methods often focus on simplified settings in discrete time (e.g., Lim et al., 2018; Bica et al., 2020; Kuzmanovic et al., 2021; Li et al., 2021; Melnychuk et al., 2022) but not in continuous time. In contrast, there is only a single neural method that operates in continuous time, namely, TE-CDE (Seedat et al., 2022). Yet, this method lacks rigorous uncertainty quantification.

|  | Continuous time | Outcome uncertainty | Model uncertainty |
|---|:---:|:---:|:---:|
| RMSNs (Lim et al., 2018) | ✗ | ✗ | ✗ |
| CRN (Bica et al., 2020) | ✗ | ✗ | ✗ |
| G-Net (Li et al., 2021) | ✗ | ✓ | (✗) |
| CT (Melnychuk et al., 2022) | ✗ | ✗ | ✗ |
| TE-CDE (Seedat et al., 2022) | ✓ | ✗ | (✗) |
| **BNCDE** (ours) | ✓ | ✓ | ✓ |

(✗): Authors use only MC dropout for model uncertainty

Table 1: Overview of key neural methods for treatment effect estimation over time. Model uncertainty (*epistemic*) refers to the uncertainty with respect to the optimal model parameters. Outcome uncertainty (*aleatoric)* is the uncertainty that is inherent to the data-generating process. While model uncertainty decreases with increasing sample size, outcome uncertainty does not.

In this work, we develop a novel neural method for treatment effect estimation from observational data in continuous time that allows for Bayesian uncertainty quantification. For this purpose, we propose the *Bayesian neural controlled differential equation* (called BNCDE). In our BNCDE, we follow the *Bayesian* paradigm to account for both model uncertainty and outcome uncertainty. To the best of our knowledge, ours is the first tailored neural method for uncertainty-aware treatment effect estimation in continuous time.

In our BNCDE, the time dimension is modeled through a coupled system of neural controlled differential equations and neural stochastic differential equations, where the neural stochastic differential equations allow for tractable variational Bayesian inference. Specifically, we use latent neural SDEs to parameterize the posterior distribution of the weights in the neural controlled differential equations. By design, the solutions to our SDEs are stochastic weight processes based on which we then compute the Bayesian posterior predictive distribution of the potential outcomes in continuous time.

We contribute to *three different streams of the literature*:[1] (1) We contribute to the *literature on treatment effect estimation*: We develop a novel method for treatment effect estimation from observational data in continuous time with uncertainty quantification. (2) We contribute the the *neural differential equation literature*: To the best of our knowledge, we are the first to propose a Bayesian version of neural controlled differential equations. (3) We contribute to the *medical literature*: We show empirically that our method yields state-of-the-art performance, paving the way for reliable, uncertainty-aware medical decision making.

## 2 RELATED WORK

We discuss methods related to (i) individualized treatment effect estimation over time and (ii) neural ordinary differential equations, as well as Bayesian methods for (i) and (ii). Thereby, we show that a Bayesian method for uncertainty quantification in our setting is missing (see Table 1). We emphasize that we focus on methods for *individualized* treatment effect estimation over time and *not* for average treatment effect estimation (e.g. Robins et al., 2000; Robins & Hernán, 2009; Rytgaard et al., 2022; Frauen et al., 2023).

**Treatment effect estimation over time:** Many works focus on estimating heterogeneous treatment effects from observational data in the *static* setting (e.g., Johansson et al., 2016; Alaa & van der Schaar, 2017; Louizos et al., 2017; Shalit et al., 2017; Yoon et al., 2018; Zhang et al., 2020; Melnychuk et al., 2023). In contrast, only a few works consider individualized treatment effect estimation in the *dynamic* setting, that is, *over time*.[2] We focus on *neural* methods for this task, as existing non-parametric methods (Xu et al., 2016; Schulam & Saria, 2017; Soleimani et al., 2017) impose strong assumptions on the outcome distribution, are not designed for multi-dimensional outcomes and static covariate data, and scalability is limited.

(1) Some methods operate in *discrete time*. Examples are the recurrent marginal structural networks (RMSNs) (Lim et al., 2018), counterfactual recurrent network (CRN) (Bica et al., 2020),

---

[1]`https://github.com/konstantinhess/Bayesian-Neural-CDE`
[2]Brouwer et al. (2022) focus on a different setting with a single, static treatment (see Supplement E).

G-Net (Li et al., 2021), and causal transformer (CT) (Melnychuk et al., 2022). However, these methods are all limited to *discrete time* (e.g., regular recordings such as in daily or hourly time steps), which is often unrealistic in medical practice.

(2) A more realistic approach is to estimate treatment effects in *continuous* time. To the best of our knowledge, the only neural method for that purpose is the treatment effect neural controlled differential equation (TE-CDE) (Seedat et al., 2022). TE-CDE leverages neural controlled differential equations (CDEs) to capture treatment effects in continuous time (see Supplement C for details on TE-CDE). However, unlike our method, TE-CDE does *not* allow for uncertainty quantification.

In the continuous time setting, the timestamps of observation may be informative about the potential outcomes and bias their estimates. A general framework to address informative sampling has been developed in (Vanderschueren et al., 2023) and applied to TE-CDE. We later also apply this approach to our BNCDE; see Supplement K.

**Uncertainty quantification for treatment effect estimation:** Jesson et al. (2020) highlight the importance of Bayesian uncertainty quantification for treatment effect estimation in the *static* setting. In the *time-varying* setting, existing methods are limited in that they use Monte Carlo (MC) dropout (Gal & Ghahramani, 2016) as an ad-hoc solution. However, MC dropout relies upon mixtures of Dirac distributions in parameter space, which leads to approximations of the true posterior which are questionable and *not* faithful (Le Folgoc et al., 2021). In contrast, a tailored neural method for *Bayesian* uncertainty quantification in the continuous time setting is still missing.

**Neural differential equations:** Neural ordinary differential equations (ODEs) can be seen as infinitely-deep residual neural networks (Chen et al., 2018; Haber & Ruthotto, 2018; Lu et al., 2018), where infinitesimal small hidden layer transformations correspond to the dynamics of a time-evolving latent vector field. Neural CDEs (Kidger et al., 2020; Morrill et al., 2021) extend neural ODEs in order to process time series data in continuous time (for more details, see Supplement D).

**Uncertainty quantification for neural ODEs:** There are only a few existing approaches for this task. (1) Variants of Markov chain Monte Carlo (MCMC) have directly been applied to neural ODEs (Dandekar et al., 2022). However, MCMC methods are known to scale poorly to high dimensions. (2) Fast approximate Bayesian inference can be achieved through Laplace approximation (Ott et al., 2023). (3) $ODE^2VAE$ (Yıldız et al., 2019) uses Gaussian distributions for variational inference in neural ODEs. Yet, both (2) and (3) rely on unimodal distributions with limited expressiveness. (4) The posterior can be approximated through neural stochastic differential equations (SDEs) (Li et al., 2020). A key benefit of this is that neural SDEs as a variational family are both scalable and arbitrarily expressive (Tzen & Raginsky, 2019; Xu et al., 2022). However, no work has integrated Bayesian uncertainty quantification into neural CDEs.

**Research gap:** As shown above, there is no method for treatment effect estimation in *continuous time* with *Bayesian* uncertainty quantification. To fill this gap, we propose our BNCDE. To the best of our knowledge, our BNCDE is also the first Bayesian neural CDE.

## 3 PROBLEM FORMULATION

**Setup:** Let $t \in [0, \bar{T}]$ be the observation time and let $\tau \in (0, \Delta]$ be the prediction window. We then consider $n$ patients (i.i.d.) as realizations of the following variables: (1) *Outcomes* over time are given by $Y_t \in \mathbb{R}$ (e.g., tumor volume). (2) *Covariates* $X_t \in \mathbb{R}^{d_x}$ include additional patient information such as comorbidity. (3) *Treatments* are given by $A_t \in \{0, 1\}^{d_a}$, where multiple treatments can be administered at the same time. The treatment assignment is controlled by a multivariate counting process $N_t \in \mathbb{N}_0^{d_a}$ with intensity $\lambda(t)$, where $N_t$ corresponds to the number of treatments assigned up to a specific point in time $t$. This is consistent with medical practice where the same treatment is oftentimes applied multiple times. For example, cancer patients may receive multiple cycles of chemotherapy (Curran et al., 2011).

Importantly, we focus on a setting in *continuous time*. That is, the outcomes and covariates of each patient $i$ are recorded at timestamps $\{t_0^i, t_1^i, \ldots, t_{m_i}^i\}$ with $t_0^i = 0$ and $t_{m_i}^i = \bar{t}^i$, where $m_i$ is the number of timestamps and $\bar{t}^i$ is the latest observation time for patient $i$.[3] Observation times

---

[3]W.l.o.g., we assume that $t_0^i = 0$ for all $i$.

are assumed to follow another, possibly history-dependent, intensity process $\zeta(t)$. This is a crucial difference from a setting in discrete time, since, in our setting, the timestamps are arbitrary and thus non-regular. Further, the timestamps may differ between patients.

We have access to observational data $h_t^i = \{y_{[0,t]}^i, x_{[0,t]}^i, a_{[0,t]}^i\}$, $t \in [0, \bar{t}^i]$, which we refer to as patient trajectories. Formally, the observed outcomes at time $t$ are given by $y_{[0,t]}^i = \bigcup_{\ell \leq k_i} \{y_{t_\ell^i}^i\}$ and the observed covariates by $x_{[0,t]}^i = \bigcup_{\ell \leq k_i} \{x_{t_\ell^i}^i\}$, where $t_{k_i}^i \leq t$ is the latest observation time up to time $t$ with $k_i \leq m_i$. In contrast, we have full knowledge of the history of assigned treatments, i.e., $a_{[0,t]}^i = \bigcup_{s \leq t} \{a_s^i\}$. We are interested in the potential outcome for a new patient $*$ at time $\bar{t}^* + \Delta$, $Y_{\bar{t}^* + \Delta}[a'_{(\bar{t}^*, \bar{t}^* + \Delta]}]$, for an arbitrary future sequence of treatments $a'_{(\bar{t}^*, \bar{t}^* + \Delta]} = \bigcup_{\bar{t}^* < \tau \leq \bar{t}^* + \Delta} a'_\tau$, given a patient's observed trajectory $h_{\bar{t}^*}^*$. For notation, we write $h_{(\bar{t}^i + \Delta)^-}^i = h_{\bar{t}^i}^i \cup \{a_{(\bar{t}^i, \bar{t}^i + \Delta]}^i\}$ when we include the factual future sequence of treatments. We write $h_{\bar{t}^i + \Delta}^i = h_{(\bar{t}^i + \Delta)^-}^i \cup \{y_{\bar{t}^i + \Delta}^i\}$ when we further include the realized observed outcome for this future sequence of treatments.

**Estimation Task:** Our objective is to predict the potential outcome for a future sequence of assigned treatments, given the observed patient history. In the following, we adopt the potential outcomes framework (Neyman, 1923; Rubin, 1978) and its extensions to the time-varying setting (Lok, 2008; Robins & Hernán, 2009; Saarela & Liu, 2016; Rytgaard et al., 2022).

Unique to our setting is that we do not focus on simple point estimates but perform *Bayesian* uncertainty quantification. In the Bayesian framework, model parameters $\omega \in \Omega$ are assigned a prior distribution $p(\omega)$. Further, for an individual $*$ and a future sequence of treatments $a'_{(\bar{t}^*, \bar{t}^* + \Delta]}$, the potential outcome $Y_{\bar{t}^* + \Delta}[a'_{(\bar{t}^*, \bar{t}^* + \Delta]}]$ has a likelihood $p(Y_{\bar{t}^* + \Delta}[a'_{(\bar{t}^*, \bar{t}^* + \Delta]}] \mid h_{\bar{t}^*}^*, \omega)$, *given* a parameter realization $\omega$ and patient trajectory $h_{\bar{t}^*}^*$. We thus aim to estimate the *posterior predictive distribution*

$$p(Y_{\bar{t}^* + \Delta}[a'_{(\bar{t}^*, \bar{t}^* + \Delta]}] \mid h_{\bar{t}^*}^*, \mathcal{H}) = \mathbb{E}_{p(\omega \mid \mathcal{H})} \left[ p(Y_{\bar{t}^* + \Delta}[a'_{(\bar{t}^*, \bar{t}^* + \Delta]}] \mid h_{\bar{t}^*}^*, \omega) \right], \tag{1}$$

which is the weighted average likelihood under the parameter posterior distribution $p(\omega \mid \mathcal{H})$ given the training data $\mathcal{H} = \bigcup_{i=1}^n h_{\bar{t}^i + \Delta}^i$. Of note, our setting is different from other works such as TE-CDE (Seedat et al., 2022), where only point estimates such as $\mathbb{E}[Y_{\bar{t}^* + \Delta}[a'_{(\bar{t}^*, \bar{t}^* + \Delta]}] \mid h_{\bar{t}^*}^*]$ are computed but *without* uncertainty quantification. Instead, we estimate the *full distribution* of the potential outcomes.

**Identifiability:** The above estimation task is challenging due to the fundamental problem of causal inference (Imbens & Rubin, 2015) in that only factual but never counterfactual patient trajectories are observed. To ensure identifiability from observational data, we make the following three assumptions that are standard in the literature (Lok, 2008; Robins & Hernán, 2009; Saarela & Liu, 2016; Rytgaard et al., 2022; Ying, 2022): (1) *Consistency:* Given a sequence of treatments $A_{[0, t+\tau]} = a_{[0, t+\tau]}$, $t \geq 0$ and $\tau \in [0, \Delta]$, the potential outcome $Y_{t+\tau}[a_{[0, t+\tau]}]$ coincides with the observed outcome $Y_{t+\tau}$. (2) *Overlap:* For any realization of patient history $H_t = h_t$, there is a positive probability of receiving treatment at any point $t$ in time. That is, the intensity process satisfies $0 < \lambda(t \mid h_t) < 1$. Similar to estimating the propensity score in the static setting (Schweisthal et al., 2023), the intensity can be estimated from data (Leemis, 1991). (3) *Unconfoundedness:* The treatment assignment probability is independent of future outcomes and unobserved information. This means that the intensity process satisfies $\lambda(t \mid h_t) = \lambda(t \mid h_t, \mathcal{F}(Y_s[a'_{(t,s]}] : s > t))$, where $\mathcal{F}(Y_s[a'_{(t,s]}] : s > t)$ is the filtration generated by future potential outcomes.

## 4 Bayesian Neural Controlled Differential Equation

### 4.1 Architecture

Our BNCDE builds upon an encoder-decoder architecture (see Fig. 1) with three components. The Ⓐ **encoder** receives the patient trajectory $H_t$ in continuous time and encodes it into a hidden representation $Z_t$ up to time $\bar{T}$. The Ⓑ **decoder** then takes the hidden representation $Z_{\bar{T}}$ together with a future sequence of treatments $a'_\tau$ and transforms it into a new hidden representation $\tilde{Z}_\tau$ for $0 < \tau \leq \Delta$, where $\Delta$ is the desired prediction window. Both the encoder and the decoder each consist of (i) a neural CDE and (ii) a latent neural SDE (see Supplement D for an overview).

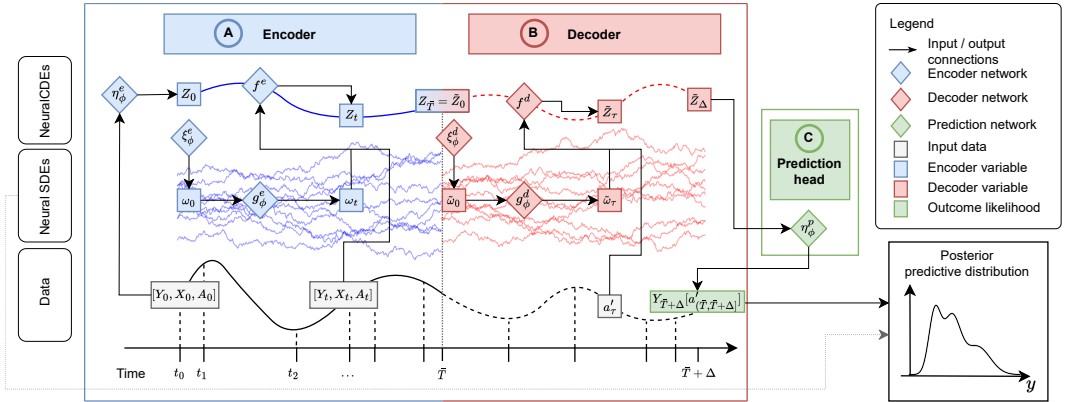

Figure 1: Our BNCDE consists of an encoder, a decoder, and a prediction head.

(i) The neural CDEs compute hidden representations of the patient trajectories in continuous time. The hidden representations in the neural CDEs allow us to model non-linearities in the data, reduce dimensionality, and capture dependencies between observed variables $(Y_t, X_t, A_t)$. (ii) The latent neural SDEs approximate the posterior distribution of the neural CDE weights. Finally, the Ⓒ **prediction head** receives $\tilde{Z}_\Delta$ and parameterizes the likelihood of the potential outcome. We explain the different components in the following.

Ⓐ **Encoder:** The encoder has two components, namely a *neural CDE* and a latent *neural SDE*: (i) The neural CDE encodes a hidden representation that is driven by the patient history. (ii) The latent neural SDE approximates the posterior distribution of the neural CDE weights.

The *neural CDE* consists of a trainable embedding network $\eta^e_\phi : \mathbb{R}^{1+d_x+d_a} \to \mathbb{R}^{d_z}$ that encodes the initial outcome $Y_0$, covariates $X_0$, and treatments $A_0$ into a hidden state $Z_0$. The hidden state $Z_0$ serves as the initial condition of the neural CDE. Formally, it is given by

$$Z_t = \int_0^t f^e(Z_s, s \mid \omega_s)\, \mathrm{d}[Y_s, X_s, A_s], \quad t \in (0, \bar{T}], \quad Z_0 = \eta^e_\phi(Y_0, X_0, A_0), \tag{2}$$

where $f^e(Z_t, t \mid \omega_t) : \mathbb{R}^{d_z+1} \to \mathbb{R}^{d_z \times (1+d_x+d_a)}$ is the neural vector field *controlled* by $[Y_t, X_t, A_t]$ *given* the network weights $\omega_t$. In particular, $f^e$ is a *Bayesian* neural network. It encodes information about the patient patient history into the hidden states $Z_t$ at any time $t \in [0, \bar{T}]$. Given the observed data, the posterior distribution of the random weights $\omega_t$ is modeled through the latent neural SDE. Hence, the weights are not static but time-varying. The output $Z_{\bar{T}}$ is then passed to the decoder.

The *neural SDE* is used to model the neural CDE weights $\omega_t \in \mathbb{R}^{d_\omega}$. Here, we assume that the weights evolve over time according to a stochastic process, where the stochastic process is the solution to the latent neural SDE. Specifically, we optimize the neural SDE to approximate the joint posterior distribution $p(\omega_{[0,\bar{T}]} \mid \mathcal{H})$ of the weights $\omega_t$ on $[0, \bar{T}]$ given the training data. We refer to the joint variational distribution as $q^e_\phi(\omega_{[0,\bar{T}]})$. Then, the neural SDE is formally defined by

$$\omega_t = \int_0^t g^e_\phi(\omega_s, s)\, \mathrm{d}s + \int_0^t \sigma\, \mathrm{d}B_s, \quad t \in (0, \bar{T}], \quad \omega_0 = \xi^e_\phi, \tag{3}$$

where $g^e_\phi : \mathbb{R}^{d_\omega+1} \to \mathbb{R}^{d_\omega}$ is the drift network, $\sigma$ is a constant diffusion coefficient, $B_t \in \mathbb{R}^{d_\omega}$ is a standard Brownian motion, and $\xi^e_\phi \sim \mathcal{N}(\nu^e_\phi, \sigma)$ is the initial condition.

Ⓑ **Decoder:** The decoder also consists of a *neural CDE* and a latent *neural SDE*. (i) The neural CDE is controlled by a future sequence of treatments. (ii) The latent neural SDE approximates the posterior distribution of the neural CDE weights.

The *neural CDE* takes the hidden representation $Z_{\bar{T}}$ from the encoder as the initial condition. The neural CDE is then given by

$$\tilde{Z}_\tau = \int_0^\tau f^d(\tilde{Z}_s, s \mid \tilde{\omega}_s)\, \mathrm{d}a'_s, \quad \tau \in (0, \Delta], \quad \tilde{Z}_0 = Z_{\bar{T}}, \tag{4}$$

where $f^d(\tilde{Z}_\tau, \tau \mid \tilde{\omega}_\tau) : \mathbb{R}^{d_z+1} \to \mathbb{R}^{d_z \times (1+d_x+d_a)}$ is the *Bayesian* neural vector field *controlled* by a future sequence of treatments $a'_\tau$ *given* the network weights $\tilde{\omega}_\tau$. The hidden state $\tilde{Z}_\Delta$ is then passed to the prediction head.

The *neural SDE* is defined in the same way as in the encoder. It is used to make a variational approximation $q_\phi^d(\tilde{\omega}_{[0,\Delta]})$ of the joint posterior distribution $p(\tilde{\omega}_{[0,\Delta]} \mid \mathcal{H}, \omega_{[0,\bar{T}]})$ of the weights $\tilde{\omega}_\tau$ on $[0,\Delta]$ given the training data and the neural CDE weights from the encoder. Formally, the neural SDE is given by

$$\tilde{\omega}_\tau = \int_0^\tau g_\phi^d(\tilde{\omega}_s, s)\, \mathrm{d}s + \int_0^\tau \sigma\, \mathrm{d}\tilde{B}_s, \quad \tau \in (0,\Delta], \quad \tilde{\omega}_0 = \xi_\phi, \tag{5}$$

where $g_\phi^d : \mathbb{R}^{d_{\tilde{\omega}}+1} \to \mathbb{R}^{d_{\tilde{\omega}}}$ is the drift network, $\tilde{B}_\tau \in \mathbb{R}^{d_{\tilde{\omega}}}$ is another standard Brownian motion, and $\xi_\phi^d \sim \mathcal{N}(\nu_\phi^d, \sigma)$ is the initial condition.

Ⓒ **Prediction head:** The prediction head is a trainable mapping $\eta_\phi^p : \mathbb{R}^{d_z} \to \mathbb{R}^2$. It takes the hidden state $\tilde{Z}_\Delta$ as input and then predicts the expected value $\mu_{\bar{T}+\Delta}$ and the outcome uncertainty $\Sigma_{\bar{T}+\Delta}$ of the potential outcome at time $\bar{T} + \Delta$ via

$$(\mu_{\bar{T}+\Delta}, \Sigma_{\bar{T}+\Delta}) = \eta_\phi^p(\tilde{Z}_\Delta). \tag{6}$$

## 4.2 Coupled System of Neural Differential Equations

We can write our BNCDE as a *coupled* system of differential equations, which allows our BNCDE to learn in an end-to-end manner. Formally, the system of neural differential equations is given by

$$\mathrm{d}\begin{pmatrix} \tilde{\omega}_\tau \\ \tilde{Z}_\tau \end{pmatrix} = \begin{pmatrix} g_\phi^d(\tilde{\omega}_\tau, \tau) \\ f^d(\tilde{Z}_\tau, \tau \mid \tilde{\omega}_\tau) \end{pmatrix} \mathrm{d}\begin{pmatrix} \tau \\ a'_\tau \end{pmatrix} + \sigma\, \mathrm{d}\begin{pmatrix} \tilde{B}_\tau \\ 0 \end{pmatrix}, \tag{7}$$

$$\mathrm{d}\begin{pmatrix} \omega_t \\ Z_t \end{pmatrix} = \begin{pmatrix} g_\phi^e(\omega_t, t) \\ f^e(Z_t, t \mid \omega_t) \end{pmatrix} \mathrm{d}\begin{pmatrix} t \\ [Y_t, X_t, A_t] \end{pmatrix} + \sigma\, \mathrm{d}\begin{pmatrix} B_t \\ 0 \end{pmatrix}, \tag{8}$$

with initial conditions

$$\tilde{Z}_0 = Z_{\bar{T}}, \quad \tilde{\omega}_0 = \xi_\phi^d, \quad Z_0 = \eta_\phi^e(Y_0, X_0, A_0) \quad \text{and} \quad \omega_0 = \xi_\phi^e, \tag{9}$$

and where $t \in (0, \bar{T}]$ and $\tau \in (0, \Delta]$. The prediction head then outputs $(\mu_{\bar{T}+\Delta}, \Sigma_{\bar{T}+\Delta}) = \eta_\phi^p(\tilde{Z}_\Delta)$. The embedding network $\eta_\phi^e$, the prediction head $\eta_\phi^p$, the initial conditions $\xi_\phi^e$ and $\xi_\phi^d$, and the drift networks $g_\phi^e$ and $g_\phi^d$ are learned via gradient-based methods.

For our model, the use of neural SDEs has three main benefits. (1) Neural SDEs allow us to approximate parameter posterior distributions that are *arbitrarily complex*. We emphasize that, although $\sigma$ is constant, prior research (Tzen & Raginsky, 2019; Xu et al., 2022) shows that given a sufficiently expressive family of drift networks, neural SDEs are able to approximate the true posterior distributions with arbitrary accuracy. (2) Our neural SDEs are *non-autonomous*, which means that they directly incorporate dependency on time. This way, we make sure that the weights learn to adjust their dynamics over time. (3) Our neural SDEs are learned through variational Bayes and enable *inference in seconds*.

## 4.3 Posterior Predictive Distribution

Our model generates the full *posterior predictive distribution* of the potential outcome, given patient history and training data. For a potential outcome at time $\bar{t}^* + \Delta$, our variational approximation of the posterior predictive distribution in Eq. 1 is given by

$$\mathbb{E}_{p(\tilde{\omega}_{[0,\Delta]}, \omega_{[0,\bar{t}^*]} \mid \mathcal{H})} \left[ p(Y_{\bar{t}^*+\Delta}[a'_{(\bar{t}^*, \bar{t}^*+\Delta]}] \mid \tilde{\omega}_{[0,\Delta]}, \omega_{[0,\bar{t}^*]}, h_{\bar{t}^*}^*) \right]$$

$$\approx \mathbb{E}_{q_\phi^e(\omega_{[0,\bar{t}^*]})} \left\{ \mathbb{E}_{q_\phi^d(\tilde{\omega}_{[0,\Delta]})} \left[ p(Y_{\bar{t}^*+\Delta}[a'_{(\bar{t}^*, \bar{t}^*+\Delta]}] \mid \tilde{\omega}_{[0,\Delta]}, \omega_{[0,\bar{t}^*]}, h_{\bar{t}^*}^*) \right] \right\}. \tag{10}$$

Hence, our posterior predictive distribution can be thought of as a Gaussian mixture model with infinitely many mixture components parameterized by the weights $\omega_{[0,\bar{T}^*]}$ and $\tilde{\omega}_{[0,\Delta]}$. We provide a formal derivation in Supplement F.

We emphasize that our model incorporates both (1) *model uncertainty* (epistemic) and (2) *outcome uncertainty* (aleatoric): (1) *Model uncertainty* is represented by the variance of the latent neural SDEs. The variances of the marginals $q_\phi^e(\omega_t)$ and $q_\phi^d(\omega_\tau)$ are larger for patient histories $h_{\bar{t}^*}^*$ and future sequences of treatments $a'_{(\bar{t}^*, \bar{t}^*+\Delta]}$ that are unlike any observations in the training data $\mathcal{H}$. Correspondingly, this leads to a larger variance of the posterior predictive distribution. Our BNCDE therefore informs about uncertainty in predictions due to lack of data support. (2) *Outcome uncertainty* is determined by the likelihood variance $\Sigma_{\bar{t}^*+\Delta}$. A larger likelihood variance implies that for a given patient history $h_{\bar{t}^*}^*$ and a future sequence of treatments $a'_{(\bar{t}^*, \bar{t}^*+\Delta]}$, there is a higher degree of variability in the potential outcome at future time $\bar{t}^* + \Delta$.

### 4.4 TRAINING

Our BNCDE is trained by maximizing the evidence lower bound (ELBO), which further requires the specification of prior distributions of the neural CDE weights and a likelihood.

**Priors:** Our chosen priors for both the encoder weights $\omega_t$ and the decoder weights $\tilde{\omega}_\tau$ are independent Ornstein-Uhlenbeck (OU) processes due to their finite variance in the time limit. We set the prior drifts to $h^e(\omega_t) = (-\omega_t)$ and $h^d(\tilde{\omega}_\tau) = (-\tilde{\omega}_\tau)$ respectively and the diffusion coefficients to $\sigma$.

**Likelihood:** The likelihood in Eq. 10 is a normal distribution parameterized by the prediction head in Eq. 6. That is, we model the likelihood as

$$Y_{\bar{T}+\Delta}[a'_{(\bar{T},\bar{T}+\Delta]}] \mid (\tilde{\omega}_{[0,\Delta]}, \omega_{[0,\bar{T}]}, H_{\bar{T}}) \sim \mathcal{N}(\mu_{\bar{T}+\Delta}, \Sigma_{\bar{T}+\Delta}). \tag{11}$$

**ELBO:** For training, we optimize the variational posterior distribution of the weights using the data $\mathcal{H}$. Specifically, we maximize the ELBO for an observation $i$ given by

$$
\begin{aligned}
\log p(y_{\bar{t}^i+\Delta}^i \mid h_{(\bar{t}^i+\Delta)^-}^i) \geq & \mathbb{E}_{q_\phi^e(\omega_{[0,\bar{t}^i]})} \Big\{ \mathbb{E}_{q_\phi^d(\tilde{\omega}_{[0,\Delta]})} \Big[ \log p(y_{\bar{t}^i+\Delta}^i \mid \tilde{\omega}_{[0,\Delta]}, \omega_{[0,\bar{t}^i]}, h_{(\bar{t}^i+\Delta)^-}^i) \Big] \Big\} \\
& - D_{\mathrm{KL}}[q_\phi^d(\tilde{\omega}_{[0,\Delta]}) \parallel p(\tilde{\omega}_{[0,\Delta]})] - D_{\mathrm{KL}}[q_\phi^e(\omega_{[0,\bar{t}^i]}) \parallel p(\omega_{[0,\bar{t}^i]})], \quad (12)
\end{aligned}
$$

where $D_{\mathrm{KL}}$ is the Kullback–Leibler divergence, $p(\tilde{\omega}_{[0,\Delta]})$ and $p(\omega_{[0,\bar{t}^i]})$ are the joint distributions of the OU priors on $[0, \Delta]$ and $[0, \bar{t}^i]$ respectively. We provide a full derivation of the ELBO loss in Supplement F. A summary of the numerical ELBO approximation is provided in Supplement G. Further implementation details of our model are provided in Supplement H.

## 5 NUMERICAL EXPERIMENTS

### 5.1 SETUP

**Data:** We benchmark our method using the established pharmacokinetic-pharmacodynamic tumor growth model by Geng et al. (2017). Variants of this model are the standard used to assess the performance of time-varying treatment effect models (Lim et al., 2018; Bica et al., 2020; Li et al., 2021; Melnychuk et al., 2022; Seedat et al., 2022; Vanderschueren et al., 2023). In the tumor growth model, the outcome $Y_t$ is the tumor volume Geng et al. (2017). At time $t$, a treatment may be applied and, if so, includes chemotherapy, radiotherapy, or both. We employ the same model variant as in Vanderschueren et al. (2023), where observations are sampled non-regularly, that is, in continuous time. We provide more details in Supplement B.

**Baselines:** Due to the novelty of our setting, there are no existing baselines for uncertainty quantification in continuous time (see Table 1). The only comparable method is **TE-CDE** with Monte Carlo (MC) dropout (Seedat et al., 2022). Implementation details are in Supplement H.

**Performance metrics:** We use different metrics to assess whether the uncertainty estimates of our method are faithful and sharp. To this end, we examine the posterior predictive *credible intervals* (CrIs) of the potential outcomes generated by the different methods. For each patient, we compute

the individual equal-tailed $(1 - \alpha)$ posterior predictive CrI with $\alpha \in [0.01, 0.05]$. This choice is motivated by medical practice, where the treatment effectiveness is typically evaluated based on the 95% and 99% CrIs. We assess the faithfulness of the CrIs by computing the *empirical coverage*, that is, the proportion of times the CrIs contain the outcomes in the test data. We further assess the sharpness of the computed CrIs by reporting the median *width* of the CrIs between 95% and 99% over all patients in the test data. We report the results for the prediction windows $\Delta = 1, 2, 3$. Results for additional prediction windows in Supplement J. We report the averaged results along with the standard deviation over five different seeds.

We further evaluate the *error in the point estimates*. Specifically, we compute the mean squared error (MSE) of the Monte Carlo mean estimates of the observed outcomes.

## 5.2 RESULTS

■ *Faithfulness:* We first evaluate whether the estimated CrIs are faithful. For this, we show the empirical coverage across different quantiles $(1 - \alpha)$. The results are in Fig. 2. For our BNCDE, the $(1 - \alpha)$ CrI almost always contains at least $(1 - \alpha)$ of the outcome, implying that the estimated CrIs are generally faithful. In contrast, the estimated CrIs for TE-CDE are generally *not* faithful. In particular, the CrIs from TE-CDE often contain fewer outcomes than the $(1 - \alpha)$ level should guarantee, implying that the uncertainty estimates from TE-CDE are overconfident. This is in line with the literature, where MC dropout is found to produce poor approximations of the posterior (Le Folgoc et al., 2021). For longer prediction windows $\Delta$, the advantages from our method are even greater. In sum, the results demonstrate that our method is clearly superior.

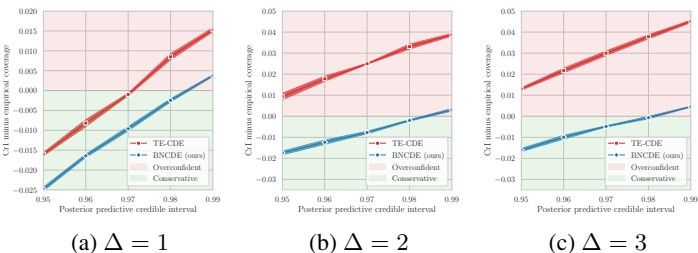

(a) $\Delta = 1$      (b) $\Delta = 2$      (c) $\Delta = 3$

Figure 2: *Faithfulness:* Empirical coverage across different CrI quantiles. Shown are different prediction windows $\Delta = 1, 2, 3$. Areas in green (red) indicate that the CrIs are faithful (not faithful).

■ *Sharpness:* We compare the median width of estimated CrIs (see Fig. 3). The CrIs from our BNCDE are significantly sharper than those of TE-CDE. This holds for all prediction windows, which further demonstrates the effectiveness of our method.

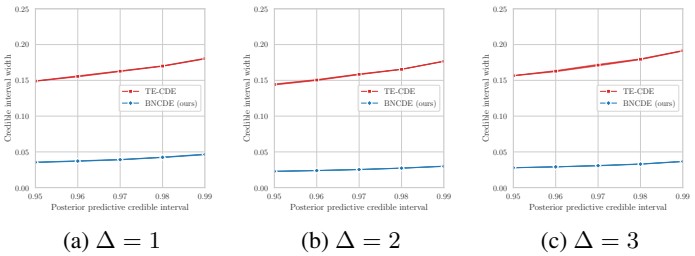

(a) $\Delta = 1$      (b) $\Delta = 2$      (c) $\Delta = 3$

Figure 3: *Sharpness:* Width of the CrIs (median) for different quantiles $\alpha$.

■ *Error in point estimates:* We further compare the errors in the Monte Carlo mean estimates of the treatment effects. The rationale is that our BNCDE may be better at computing the posterior distribution, yet the more complex architecture could hypothetically let the point estimates deteriorate. However, this is not the case, and we see that our BNCDE is clearly superior (see Fig. 4). We further find that the performance gain from our method is robust across different levels of noise in the data-generating process (i.e., larger $\text{Var}(\epsilon_t)$).

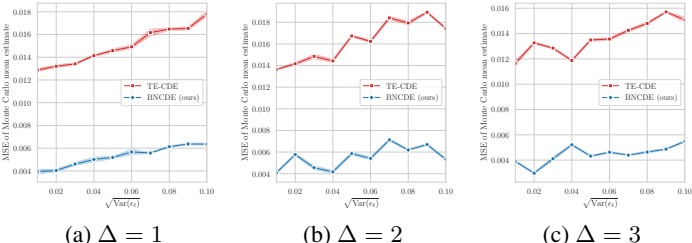

| (a) $\Delta = 1$ | (b) $\Delta = 2$ | (c) $\Delta = 3$ |

Figure 4: *Error in point estimates:* Reported is the median over the mean squared errors (MSE) of the point estimates of the outcomes. The results are based on test data that is generated with varying levels of noise, i.e., $\text{Var}(\epsilon_t)$.

### 5.3 COMPARISON OF MODEL UNCERTAINTY

We now provide a deep-dive comparing the uncertainty estimates for model uncertainty only (and thus without outcome uncertainty). We do so for two reasons: (1) We can directly compare the model uncertainty from both our BNCDE and TE-CDE with MC dropout as the latter is limited to model uncertainty. (2) We can better understand the role of the neural SDE in our method (i.e., before the variational approximations are passed to the prediction head). For TE-CDE, we thus use the variance in the MC dropout samples as a measure of model uncertainty. For our BNCDE, we capture the model uncertainty with the Monte Carlo variance in the means of the likelihood $\mu_{\bar{t}^*+\Delta}$ under the neural SDE weights $\omega_{[0,\bar{t}^*]}$ and $\tilde{\omega}_{[0,\Delta]}$. However, both measures are not directly comparable. Hence, inspired by analyses for static settings (Jesson et al., 2020; 2021; Oprescu et al., 2023), we compare the *deferral rate*. That is, we report errors in the MSE of the treatment effect as we successively withhold samples with the largest Monte Carlo variance.

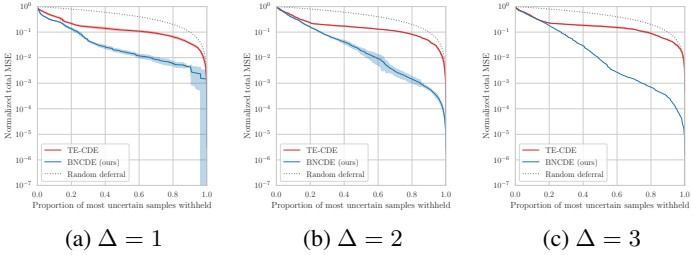

| (a) $\Delta = 1$ | (b) $\Delta = 2$ | (c) $\Delta = 3$ |

Figure 5: To compare model uncertainty, the normalized MSE of estimated treatment effects between the treatment arms versus deferral rate is shown.

Fig. 5 shows the normalized MSE across different proportions of withheld samples. For TE-CDE with MC dropout, the MSE decreases almost randomly, implying that the MC estimates as a measure of model uncertainty are highly uninformative (and close to random deferral). For our BNCDE, a larger MSE in the treatment effect estimates corresponds to a higher model uncertainty, thus implying that the model uncertainty in our method is more informative.

**Extension:** We repeat our experiments for informative sampling (see Supplement K). For this, we extend both our BNCDE and TE-CDE using the inverse intensity weighting method from Vanderschueren et al. (2023). We find that our method remains clearly superior.

**Conclusion:** Our results show the following: (1) Our BNCDE produces posterior predictive CrIs that are *faithful*. In contrast, the CrIs from TE-CDE can be overconfident, which, in medicine, may lead to treatment decisions that are dangerous. (2) Our BNCDE further provides approximations of the CrIs that are *sharper*. (3) Our BNCDE generates more *informative* estimates of model uncertainty compared to TE-CDE with MC dropout. (4) Our BNCDE is further more *robust* against noise than TE-CDE with MC dropout and, also, (5) highly effective for longer prediction windows. We discuss the applicability of our BNCDE in medical scenarios in Supplement A.

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

## A  DISCUSSION ON THE APPLICABILITY IN MEDICAL SCENARIOS

In our main paper, we demonstrated that BNCDE represents a significant step toward reliable, uncertainty-aware decision-making in medicine. To this end, we improve over existing methods by leveraging the Bayesian paradigm and by introducing a novel neural approach for uncertainty-aware treatment effect estimation. Specifically, we provide strong empirical evidence that our BNCDE based on a combination of neural CDEs and latent neural SDEs outperforms the existing ad-hoc baselines, namely TE-CDE (Seedat et al., 2022) with MC dropout (Gal & Ghahramani, 2016). Additionally, we demonstrate numerically that our method is compatible with other extensions, such as the informative sampling framework by Vanderschueren et al. (2023) and balanced representations (Bica et al., 2020; Melnychuk et al., 2022; Seedat et al., 2022) (see Supplements K and L).

Our experiments further demonstrate that our method holds great relevance in clinical settings. First, our method allows for treatment effect estimation in *continuous time*. That is, it does not rely on the unrealistic assumption of regular recording times in electronic health records (Özyurt et al., 2021), but can instead account for highly irregularly sampled recording times. In electronic health records, irregular observation times are common and quasi-standard (Alaa et al., 2017; Allam et al., 2021). Because of this, the assumptions in existing methods for discrete time are directly *violated*, while a particular strength of our method is that we explicitly account for irregular sampling. Second, our method does not only provide point estimates of the potential outcomes but estimates the *full posterior predictive distribution*. This is crucial in personalized medicine, as physicians may want to base their decisions on uncertainty estimates. In fact, uncertainty estimates are imperative for all medical applications to avoid patient harm (Banerji et al., 2023). To account for that, our posterior predictive distribution provides insights into both (i) model uncertainty (see Fig. 5) and (ii) outcome uncertainty (see Supplement I) of the potential outcomes in continuous time, and is the first tailored neural method for this task.

Combining outcome estimation in continuous time and rigorous uncertainty quantification in a single, neural end-to-end architecture is a major improvement over all existing neural baselines (see Section 2). Furthermore, existing non-parametric approaches (Xu et al., 2016; Schulam & Saria, 2017; Soleimani et al., 2017) impose strong assumptions on the outcome distribution, they are not designed for multi-dimensional outcomes and static covariate data, and they face scalability issues. For instance, Gaussian Process methods as in (Schulam & Saria, 2017) require a matrix inversion of the covariance matrix involving *all* patients in the training data. Due to the cubic scaling of this operation, this is completely impractical in medical scenarios with constrained time and computational resources.

However, while our BNCDE presents a first and important step toward uncertainty-aware medical decision-making with neural networks, our method also has limitations as any other. We discuss them in the following:

(i) Our BNCDE remains a black-box method that is not directly explainable. While explainability is often desired in medical scenarios for reliable decision-making, this issue is not unique to our method but applies to all previously mentioned neural approaches (Lim et al., 2018; Bica et al., 2020; Li et al., 2021; Melnychuk et al., 2022; Seedat et al., 2022). As such, we suggest a careful use of our method in high-stakes decisions. Instead, we recommend the use of our method in automated or routine tasks. For example, there is a growing number of smartphone apps with digital health interventions (Murray et al., 2016), which could benefit from personalization through our method. Further, we argue that recognizing highly non-linear, complex, and inexplicable patterns in data is what neural networks are designed for; this, of course, contradicts direct explainability.

(ii) As reliability is of high importance, we emphasize that tuning the diffusion hyperparameter has noticeable impact on the approximation quality of the latent neural SDEs. In particular, during our experimental studies, increasing the diffusion constant directly led to stronger conservatism in the predictive predictive estimates.

(iii) Training our BNCDE involves solving high-dimensional stochastic differential equations. This is a numerically challenging task and is the computational bottleneck of our method. However, we emphasize that, as validated in Supplement M, the training time scales only *linearly* with the dimension of the latent neural SDEs. This is not different from the scaling of any other fully connected neural network. Furthermore, we emphasize that, once the network is trained, inference for new

patients can be achieved *in seconds*. Hence, this makes our method suitable for deployment as a decision support tool in clinical settings.

(iv) Our work focuses on approximate Bayesian inference for neural CDEs of moderate size as encountered in medical practice and has not been designed for very high-dimensional settings (e.g., inference on neural CDEs with billions of parameters). While such high-dimensional settings are out of scope for this work, uncertainty quantification is typically needed for moderately sized (rather than very large) datasets and neural networks in practice. The reason is that, for moderately sized datasets, we cannot expect to learn the true data-generating mechanisms. Instead, we expect that there is uncertainty in how medical treatments affect patient health and, to deal with this, it is imperative in medicine to make treatment choices by incorporating the underlying uncertainty (Banerji et al., 2023).

(v) Due to the highly complex structure in electronic health records, sufficient data is required to obtain robust estimation results. However, we are optimistic that this will not be an issue in the near future due to the increasing amount of available data and data sharing agreements (e.g., Johnson et al., 2016; Pollard et al., 2018; Hyland et al., 2020; Thoral et al., 2021; Rodemund et al., 2023).

(vi) As with any other method for treatment effect estimation, ours relies upon mathematical assumptions. Importantly, the above implementation of our BNCDE does not account for confounding bias. Confounding bias distorts the true causal relationship between a treatment and an outcome, stemming from the influence of an unaccounted-for variable. This occurs when a variable is associated with both the treatment and the outcome, introducing a potential source of bias. For instance, patients with more severe health conditions may be more likely to receive treatment A, while those with better health status may be given treatment B (Vokinger et al., 2021). Consequently, when estimating the effect of treatment A from data, bias towards a worse outcome emerges due to the association between health condition and treatment assignment. Addressing confounding bias in discrete-time literature involves adjustments like inverse propensity weighting (IPW) (RMSNs (Lim et al., 2018)) and G-computation (G-Net (Li et al., 2021)). However, these adjustments lead to very high estimation variance. Other neural methods such as (Bica et al., 2020; Melnychuk et al., 2022; Seedat et al., 2022) try to reduce confounding bias through balanced representations. However, balancing is an approach for *reducing estimation variance* and *not for mitigating confounding bias* (Shalit et al., 2017). Furthermore, it may even introduce an infinite data bias (Melnychuk et al., 2024) (see Supplement N for a detailed discussion on balanced representations). As in (Vanderschueren et al., 2023), we therefore decided against balanced representations in our main paper. Of note, our BNCDE is generally compatible with balancing, as we show in Supplement L. Notwithstanding, while estimation bias due to confounding is still an open issue, we emphasize that this is not unique to our method but a general problem in the literature for time-varying treatment effect estimation.

Finally, we acknowledge that treatment effect estimation in the continuous-time setting is challenging and remains subject to ongoing research. Our BNCDE faces the same lack of proper adjustments as the only other neural baseline (Seedat et al., 2022) and does not offer unbiasedness guarantees. Nevertheless, we believe that our BNCDE is a major improvement over existing methods and a direct step towards reliable decision-making in medicine.

## B  CANCER SIMULATION

The tumor data was simulated according to the lung cancer model by Geng et al. (2017), which has previously been used in (Lim et al., 2018; Bica et al., 2020; Li et al., 2021; Melnychuk et al., 2022; Seedat et al., 2022; Vanderschueren et al., 2023). In particular, we adopt the simulation setting by Vanderschueren et al. (2023), which includes irregularly sampled observations and avoids introducing confounding bias in the treatment assignment mechanism.

The tumor volume, which we refer to as the outcome variable in the main paper, evolves according to an ordinary differential equation:

$$dY_t = \left[ 1 + \underbrace{\rho \log\left(\frac{K}{Y_t}\right)}_{\text{Tumor growth}} - \underbrace{\alpha_c c_t}_{\text{Chemotherapy}} - \underbrace{(\alpha_r d_t + \beta_r d_t^2)}_{\text{Radiotherapy}} + \underbrace{\epsilon_t}_{\text{Noise}} \right] Y_t \, dt, \tag{13}$$

where $\rho$ is a growth parameter, $K$ is the carrying capacity, $\alpha_c$, $\alpha_r$ and $\beta_r$ control the chemo and radio cell kill, respectively, and $\epsilon_t$ introduces randomness into the growth dynamics. The parameters were sampled according to Geng et al. (2017) and are summarized in Table 2. The variables $c_t$ and $d_t$ are set following the work by Lim et al. (2018); Bica et al. (2020); Seedat et al. (2022). The time $t$ is measured in days.

|  | Variable | Parameter | Distribution | Value $(\mu, \sigma^2)$ |
|---|---|---|---|---|
| Tumor growth | Growth parameter | $\rho$ | Normal | $(7.00 \times 10^{-5}, 7.23 \times 10^{-3})$ |
|  | Carrying capacity | $K$ | Constant | 30 |
| Radiotherapy | Radio cell kill | $\alpha_r$ | Normal | $(0.0398, 0.168)$ |
|  | Radio cell kill | $\beta_r$ | – | Set to $\beta_r = 10 \times \alpha_r$ |
| Chemotherapy | Chemo cell kill | $\alpha_c$ | Normal | $(0.028, 7.00 \times 10^{-4})$ |
| Noise | – | $\epsilon_t$ | Normal | $(0, 0.01^2)$ |

Table 2: Sampling details of parameters used in the tumor simulation model.

We make the following, additional adjustments as informed by prior literature:

- As in (Lim et al., 2018; Bica et al., 2020; Seedat et al., 2022; Vanderschueren et al., 2023), heterogeneity between patients is introduced by modeling different subgroups. Each subgroup differs in their average response to treatment, expressed by the mean values of the normal distributions. For subgroup 1, we increase $\mu(\alpha_r)$ by 10%, and, for subgroup 2, we increase $\mu(\alpha_c)$ by 10%.

- Following Kidger et al. (2020), we add a multivariate counting variable that counts how often each treatment has been administered up to a specific time.

- As in (Vanderschueren et al., 2023), the observation process is governed by an intensity process with history-dependent intensity

$$\zeta_t^i = \text{sigmoid}\left[ \gamma \left( \frac{\bar{D}_t^i}{D} - \frac{1}{2} \right) \right], \tag{14}$$

where $\gamma$ controls the sampling informativeness, $D = 13$ cm, and $\bar{D}_t^i$ is the average tumor diameter over the last 15 days. For our informative sampling experiments in Section 5 and for the additional prediction windows in Supplement J, we set $\gamma = 1$. For our experiments in Supplement K, we increased the sampling informativeness to $\gamma = 2$.

Treatments are assigned according to either a concurrent or a sequential treatment arm (Curran et al., 2011). That is, patients receive treatment either (i) weekly chemotherapy for five weeks and then radiotherapy for another five weeks or (ii) biweekly chemotherapy *and* radiotherapy for ten weeks. Patients are randomly divided between the two treatment regimes. For any patient in the test data, we simulate both the factual outcome under the assigned treatment arm and the counterfactual outcome under the unassigned treatment arm.

For training, validation, and testing, our observed time window is 55 days with an additional prediction window of up to 5 days. For training and validation, we simulated $10,000$ and $1000$ observations, respectively. For testing, we simulated the trajectories and the outcomes for $10,000$ patients under both treatment arms, respectively. We standardized our data with the training set.

## C  TE-CDE

In this section, we give a brief overview of the so-called *treatment effect neural controlled differential equation* (TE-CDE) (Seedat et al., 2022). We present the main ideas of the original TE-CDE model but refer to the original paper for more details. We build upon our notation to avoid ambiguities.

TE-CDE uses neural controlled differential equations to predict potential outcomes in continuous time. For this, TE-CDE assumes that a hidden state variable $Z_t \in \mathbb{R}^{d_z}$ is driven by the outcome history, covariate history, and treatment history. The control path is interpolated in continuous time from observational, irregularly sampled data. The hidden state variable $Z_t$ encodes information about the potential outcomes for a given treatment assignment. In (Seedat et al., 2022), TE-CDE is trained with an adversarial loss term to enforce balanced representations (see Supplement N for a discussion on balanced representations). However, balanced representations are not the focus of our paper, and we thus omit them for better comparability.

As is common for neural CDEs, TE-CDE encodes the initial outcome variable $Y_0$, covariates $X_0$ and treatment $A_0$ into the hidden state $Z_0$ through an embedding network $\eta_\phi^e : \mathbb{R}^{1+d_x+d_a} \to \mathbb{R}^{d_z}$. The hidden state then serves as the initial condition of the controlled differential equation given by

$$Z_t = \int_0^t f_\phi^e(Z_s)\, \mathrm{d}[Y_s, X_s, A_s], \quad t \in (0, \bar{T}] \quad \text{and} \quad Z_0 = \eta_\phi^e(Y_0, X_0, A_0). \tag{15}$$

Eq. 15 is the encoder network of TE-CDE. The last hidden state $Z_{\bar{T}}$ is then passed through a decoder to predict the potential outcome $Y_{\bar{T}+\Delta}[a'_{(\bar{T}, \bar{T}+\Delta]}]$ for an arbitrary but fixed time window $\Delta$ in the future. The decoder is given by the neural CDE

$$Y_{\bar{T}+\Delta}[a'_{(\bar{T}, \bar{T}+\Delta]}] \approx \eta_\phi^d(\tilde{Z}_\Delta), \quad \tilde{Z}_\tau = \int_0^\tau f_\phi^d(\tilde{Z}_s)\, \mathrm{d}a'_s, \quad \tau \in (0, \Delta] \quad \text{and} \quad \tilde{Z}_0 = Z_{\bar{T}}, \tag{16}$$

where $\eta_\phi^d$ is the read-out mapping. In particular, for a new patient $*$, TE-CDE seeks to approximate the expected value of the potential outcome, given the patient trajectory $H_{\bar{t}^*}^*$. That is, TE-CDE approximates the quantity

$$\mathbb{E}[Y_{\bar{t}^*+\Delta}[a'_{(\bar{t}^*, \bar{t}^*+\Delta]}] \mid H_{\bar{t}^*}^*]. \tag{17}$$

Reassuringly, we highlight the following **key differences between TE-CDE and our BNCDE**:

1. TE-CDE targets Eq. 17 and thus makes a *point estimate* of the potential outcome at time $\bar{t}^* + \Delta$. In contrast, our BNCDE estimates the *full posterior predictive distribution* of the potential outcome.

2. TE-CDE *directly* optimizes the neural vector fields $f_\phi^e$ and $f_\phi^d$ of the CDE. Instead, our BNCDE parameterizes the distribution of the neural CDE weights with latent neural SDEs. This neural SDE is optimized to shape the *posterior distribution* of the neural CDE weights given the training data. As such, the architectures of TE-CDE and our BNCDE are vastly different.

3. TE-CDE may employ *MC dropout* to approximate model uncertainty. However, as we show in our paper, the tailored *neural SDEs* provide more informative model uncertainty quantification.

4. TE-CDE does *not* provide an estimate of the *outcome uncertainty quantification*. On the other hand, our BNCDE *incorporates outcome uncertainty* through the likelihood variance $\Sigma_{\bar{t}^*+\Delta}$ (see Supplement I).

5. The training objective of TE-CDE is against the *mean squared error* (MSE). In contrast, our method optimizes the *evidence lower bound* (ELBO). The ELBO seeks to balance the expected likelihood under the posterior distribution with the Kullback-Leibler divergence between prior and posterior SDEs. Therefore, our loss naturally incorporates weight regularization.

# D    NEURAL DIFFERENTIAL EQUATIONS

We provide a brief overview of (1) neural ODEs, (2) neural CDEs, and (3) neural SDEs.

**Neural ODEs:** Neural ordinary differential equations (ODEs) (Haber & Ruthotto, 2018; Lu et al., 2018) combine neural networks with ordinary differential equations. Recall that, in a traditional residual neural network with $t = 1, \ldots, \bar{T}$ residual layers, the hidden states are defined as

$$Z_t = Z_{t-1} + f_{\phi_t}(Z_{t-1}), \tag{18}$$

where $f_{\phi_t}(\cdot)$ is the $t$-th residual layer with trainable parameters $\phi_t$ and $Z_0 = X$ is the input data. These update rules correspond to an Euler discretization of a continuous transformation. That is, stacking infinitely many infinitesimal small residual transformations, $f_\phi(\cdot)$ defines a vector field induced by the initial value problem

$$Z_t = \int_0^t f_\phi(Z_s, s) \, \mathrm{d}s, \quad Z_0 = X. \tag{19}$$

Instead of specifying discrete layers and their parameters, a neural ODE describes the continuous changes of the hidden states over an imaginary time scale. A neural ODE thus learns a continuous flow of transformations: an input $X = Z_0$ is passed through an ODE solver to produce an output $\hat{Y} = Z_{\bar{T}}$. The vector field $f_\phi$ is then updated either by propagating the error backward through the solver or using the adjoint method (Chen et al., 2018).

**Neural CDEs:** Neural controlled differential equations (CDEs) (Kidger et al., 2020) extend the above architecture in order to handle time series data in *continuous time*. In a neural ODE, all information has to be captured in the input $Z_0 = X$, i.e., at time $t = 0$. Neural CDEs, on the other hand, are able to process time series data. They can thus be seen as the continuous-time analog of recurrent neural networks. For a path of covariate data $X_t \in \mathbb{R}^{d_x}$, $t \in [0, \bar{T}]$, a neural CDE is given by an embedding network $\eta_\phi^0(\cdot)$, a readout mapping $\eta_\phi^1(\cdot)$, and a neural vector field $f_\phi$ that satisfy

$$\hat{Y} = \eta_\phi^1(Z_{\bar{T}}), \quad Z_t = \int_0^t f_\phi(Z_s, s) \, \mathrm{d}X_s, \quad t \in (0, \bar{T}] \quad \text{and} \quad Z_0 = \eta_\phi^0(X_0), \tag{20}$$

where $Z_t \in \mathbb{R}^{d_z}$ and $f_\phi(Z_t, t) \in \mathbb{R}^{d_z \times d_x}$. The integral is a Riemann-Stieltjes integral, where $f_\phi(Z_s, s) \, \mathrm{d}X_s$ refers to matrix-vector multiplication. Here, the neural differential equation is said to be *controlled* by $X_t$. Under mild regularity conditions, it can be computed as

$$Z_t = \int_0^t f_\phi(Z_s, s) \frac{\mathrm{d}X_s}{\mathrm{d}s} \, \mathrm{d}s, \quad t \in (0, \bar{T}]. \tag{21}$$

Computing the derivative with respect to time requires a representation of the data stream $X_t$ for any $t \in [0, \bar{T}]$. Hence, irregularly sampled observations $((t_0, x_0), (t_1, x_1), \ldots, (t_n, x_n) = (\bar{t}, x_n))$ need to be interpolated over time, yielding a continuous time representation $X_t$. The neural vector field $f_\phi$, the embedding network $\eta_\phi^0$, and the readout network $\eta_\phi^1$ are then optimized analogous to the neural ODE. For data interpolation, Morrill et al. (2021) suggest different interpolation schemes for different tasks such as rectilinear interpolation for online prediction.

**Neural SDEs:** Neural stochastic differential equations (SDEs) (Li et al., 2020) learn stochastic differential equations from data. A neural SDE is given by a drift network $g_\phi$ and a diffusion network $\sigma_\phi$ that satisfy

$$\omega_t = \int_0^t g_\phi(\omega_s, s) \, \mathrm{d}s + \int_0^t \sigma_\phi(\omega_s, s) \, \mathrm{d}B_s, \quad t > 0 \quad \text{and} \quad \omega_0 = \xi_\phi, \tag{22}$$

where $B_t$ is a standard Brownian motion and $\xi_\phi \sim \mathcal{N}(\nu_\phi, \sigma_\phi)$ is the initial condition.

For two SDEs with shared diffusion coefficient, their Kullback-Leibler (KL) divergence can be computed on path space. This makes neural SDEs particularly suited for variational Bayesian inference. Let $g_\phi^q$ and $h$ be the drifts of the variational neural SDE and the prior SDE respectively. Let furthermore $\sigma(\omega_t, t)$ be their shared diffusion coefficient. Tzen & Raginsky (2019); Li et al. (2020) show that the KL-divergence on path space then satisfies

$$D_{\mathrm{KL}}[q_\phi^q(\omega_{[0,t]}) \, \| \, p(\omega_{[0,t]})] = \mathbb{E}_{q_\phi^q(\omega_{[0,t]})} \left[ \int_0^t \left( (g_\phi^q(\omega_s, s) - h(\omega_s, s))/\sigma(\omega_s, s) \right)^2 \, \mathrm{d}s \right]. \tag{23}$$

Of note, even for a constant shared diffusion parameter $\sigma(\omega_t, t) = \sigma$, a variational posterior distribution parameterized this way can approximate the true posterior arbitrarily closely using a sufficiently expressive family of drift functions $g_\phi^q$ (Tzen & Raginsky, 2019; Xu et al., 2022).

# E   MOTIVATION FOR USING NEURAL CDES (AND NOT NEURAL ODES)

In the following, we provide a discussion on why we used *neural CDEs* instead of neural ODEs to capture patient observations and to model future sequences of treatments for personalized decision-making. We also highlight the main differences to CF-ODE (Brouwer et al., 2022), which builds on neural ODEs to estimate treatment effects in a different setting than ours.

*Why did we opt for neural CDEs instead of neural ODEs?*

Neural ODEs can be thought of as residual neural networks with infinitely many, infinitesimal small hidden layer transformations (see SupplementD). Thus, neural ODEs are designed to describe how a deterministic system evolves, given its initial conditions (e.g., a patient's initial health condition, an initial treatment decision). However, it is impossible to change these dynamics over time, once the neural ODE is learned. This is different from neural CDEs, which adjust for sequentially incoming information (Kidger et al., 2020; Morrill et al., 2021). Further, neural CDEs make use of latent representations, which are able to capture non-linearities and complex dependencies in the data.

*Why is a single neural ODE not suitable for our task?*

There are clear theoretical reasons why using a single neural ODE to model the dynamics of $(Y_t, X_t, A_t)$ is **not** suitable. If we used a neural ODE to directly model the dynamics of $(Y_t, X_t, A_t)$, this would mean that we learn an ODE that describes the deterministic evolution of $(Y_t, X_t, A_t)$, given the initial conditions $(Y_0, X_0, A_0)$. In particular, we would then make the following assumptions that would conflict with our task:

- (i) The health conditions of a patient are completely captured in her *initial* state $(Y_0, X_0, A_0)$. The initial state is the only variable that influences the evolution of a (neural) ODE. However, we want the patient encoding to be updated over time.

- (ii) Given these initial conditions, the outcome and covariate evolution are *deterministic*. It is not possible to "correct" an ODE at a future point in time to account for the randomness of future observations. However, we want to take into account the information from the observations in the future when they are measured.

- (iii) The treatment assignment plan is *fixed* a priori and cannot be changed. That means, we would assume that sequences of treatments evolve like a deterministic process. This makes modeling of arbitrary future sequences of treatments impossible.

- (iv) Future observations cannot change the dynamics of the system and, importantly, *cannot update beliefs* about the potential outcomes. In particular, future observations cannot correct the posterior distribution that captures the uncertainty. However, we intend to update the posterior distribution when patient information is recorded at later points in time.

*How can our architecture based on neural CDEs address the above issues?*

Our architecture solves all of the above issues. The reason is that neural CDEs have much larger flexibility that is beneficial for our task:

- (i) Neural CDEs allow for *patient personalization at any point in time $t$*, and do not assume that all information is captured at time $t = 0$.

- (ii) Future observations $(Y_t, X_t, A_t)$ can *update the neural CDE*. This is crucial, because not all patient trajectories follow the same deterministic evolution.

- (iii) Using a neural CDE in our decoder, we can model *arbitrary future sequences of multiple treatments*.

- (iv) Incoming observations after time $t = 0$ can *update our beliefs*. This makes the neural CDEs particularly suited for the Bayesian paradigm.

Importantly, the hidden states $Z_t$ in a neural CDE also allow us to model non-linearities in the data, reduce dimensionality, and capture dependencies between observed variables $(X_t, Y_t, A_t)$. In summary, using neural ODEs to directly model the dynamics of $(Y_t, X_t, A_t)$ has significantly lower modeling capacities, limits patient personalization, and imposes prohibitive assumptions on the real world. To this end, we argue that it is difficult – if not impossible – to adequately model the input

through a joint neural ODE in our task, and, as a remedy, we opted for an approach based on neural CDEs instead.

Of note, the Counterfactual ODE (CF-ODE) (Brouwer et al., 2022) leverages neural ODEs for treatment-effect estimation in a different setting to ours. In the following, we demonstrate why CF-ODE is **not** applicable to our setting. We highlight the limitations of this method and clarify important differences to our BNCDE:

- (i) CF-ODE is inherently designed to forecast the treatment effect over time of a *single, static* treatment. In contrast, our BNCDE is designed to estimate treatment effects for a *sequence of multiple* treatments in the future.

- (ii) Accordingly, CF-ODE builds upon a static identifiability framework. Here, adjusting for the measurements of the patient history at the time of treatment is assumed to be sufficient to adjust for all confounders. However, the static identifiability form CF-ODE does not hold true for time-varying settings as ours.

- (iii) CF-ODE directly models the latent states in the decoder with a neural SDE. In contrast, our neural SDEs model the distribution of the neural CDE weights.

- (iv) All patient information in CF-ODE is contained *deterministically* in the initial state of the ODE. Hence, all *uncertainty* needs to be incorporated in a *deterministic*, initial state. On the other hand, our BNCDE captures patient information in the random variable $Z_{\bar{T}} = \tilde{Z}_0$. Here, the Monte Carlo variance is propagated through the encoded trajectory $Z_{[0,\bar{T}]}$, accounting for uncertainty at each point in time $t \in [0, \bar{T}]$. Therefore, model uncertainty is directly incorporated into the patient encoding.

- (iv) The use of a gated recurrent unit Cho et al. (2014) in the CF-ODE encoder implies *regular sampling* of the patient trajectories. This assumption is a *crucial violation of medical reality*, where it is standard that measurements are taken at *irregular* points in time. Therefore, CF-ODE does *not* account for uncertainty due to irregular sampling.

- (vi) The single treatment decision in CF-ODE is limited to binary treatments. Instead, our BNCDE can deal with any treatment.

# F    VARIATIONAL APPROXIMATIONS

We provide a detailed derivation of (1) our approximate posterior predictive distribution and (2) our evidence lower bound.

**Notation:** We slightly abuse notation for better readability. We denote the expectation with respect to the stochastic process $\omega_t$ on path space as

$$\int (\cdot)\, \mathrm{d}p(\omega_{[0,\bar{T}]}) = \mathbb{E}_{p(\omega_{[0,\bar{T}]})}[(\cdot)]. \tag{24}$$

The analogue applies for $\tilde{\omega}_\tau$.

**Approximate posterior predictive distribution:** In the Bayesian framework, the posterior predictive distribution of the outcome of interest $Y_{\bar{t}^*+\Delta}[a'_{(\bar{t}^*,\bar{t}^*+\Delta]}]$ given observed inputs $h^*_{\bar{t}^*}$ and training data $\mathcal{H}$ is the expectation of the outcome likelihood under the weight posterior distribution. Hence, we approximate the posterior predictive distribution via

$$p(Y_{\bar{t}^*+\Delta}[a'_{(\bar{t}^*,\bar{t}^*+\Delta]}] \mid h^*_{\bar{t}^*}, \mathcal{H}) \tag{25}$$

$$= \int p(Y_{\bar{t}^*+\Delta}[a'_{(\bar{t}^*,\bar{t}^*+\Delta]}] \mid h^*_{\bar{t}^*}, \tilde{\omega}_{[0,\Delta]}, \omega_{[0,\bar{t}^*]})\, \mathrm{d}p(\tilde{\omega}_{[0,\Delta]}, \omega_{[0,\bar{t}^*]} \mid \mathcal{H}) \tag{26}$$

$$= \int p(Y_{\bar{t}^*+\Delta}[a'_{(\bar{t}^*,\bar{t}^*+\Delta]}] \mid h^*_{\bar{t}^*}, \tilde{\omega}_{[0,\Delta]}, \omega_{[0,\bar{t}^*]})\, \mathrm{d}p(\tilde{\omega}_{[0,\Delta]} \mid \omega_{[0,\bar{t}^*]}, \mathcal{H})\, \mathrm{d}p(\omega_{[0,\bar{t}^*]} \mid \mathcal{H}) \tag{27}$$

$$\approx \int p(Y_{\bar{t}^*+\Delta}[a'_{(\bar{t}^*,\bar{t}^*+\Delta]}] \mid h^*_{\bar{t}^*}, \tilde{\omega}_{[0,\Delta]}, \omega_{[0,\bar{t}^*]})\, \mathrm{d}q^d_\phi(\tilde{\omega}_{[0,\Delta]})\, \mathrm{d}q^e_\phi(\omega_{[0,\bar{t}^*]}) \tag{28}$$

$$= \mathbb{E}_{q^e_\phi(\omega_{[0,\bar{t}^*]})}\left[\mathbb{E}_{q^d_\phi(\tilde{\omega}_{[0,\Delta]})}\left[p(Y_{\bar{t}^*+\Delta}[a'_{(\bar{t}^*,\bar{t}^*+\Delta]}] \mid \tilde{\omega}_{[0,\Delta]}, \omega_{[0,\bar{t}^*]}, h^*_{\bar{t}^*})\right]\right]. \tag{29}$$

**Evidence lower bound:** Recall that, in our notation, a training sample is $h^i_{\bar{t}^i+\Delta} = h^i_{(\bar{t}^i+\Delta)^-} \cup \{y^i_{\bar{t}^i+\Delta}\}$, where $h^i_{(\bar{t}^i+\Delta)^-}$ is the training input and $y^i_{\bar{t}^i+\Delta}$ the target. Our objective is an ELBO maximization, which we derive via

$$\log p(y^i_{\bar{t}^i+\Delta} \mid h^i_{(\bar{t}^i+\Delta)^-}) \tag{30}$$

$$= \log \int p(y^i_{\bar{t}^i+\Delta} \mid h^i_{(\bar{t}^i+\Delta)^-}, \tilde{\omega}_{[0,\Delta]}, \omega_{[0,\bar{t}^i]})\, \mathrm{d}p(\tilde{\omega}_{[0,\Delta]}, \omega_{[0,\bar{t}^i]}) \tag{31}$$

$$= \log \int p(y^i_{\bar{t}^i+\Delta} \mid h^i_{(\bar{t}^i+\Delta)^-}, \tilde{\omega}_{[0,\Delta]}, \omega_{[0,\bar{t}^i]})\, \mathrm{d}p(\tilde{\omega}_{[0,\Delta]})\, \mathrm{d}p(\omega_{[0,\bar{t}^i]}) \tag{32}$$

$$= \log \int p(y^i_{\bar{t}^i+\Delta} \mid h^i_{(\bar{t}^i+\Delta)^-}, \tilde{\omega}_{[0,\Delta]}, \omega_{[0,\bar{t}^i]})\left(\frac{q^d_\phi(\tilde{\omega}_{[0,\Delta]})q^e_\phi(\omega_{[0,\bar{t}^i]})}{p(\tilde{\omega}_{[0,\Delta]})p(\omega_{[0,\bar{t}^i]})}\right)^{-1} \mathrm{d}q^d_\phi(\tilde{\omega}_{[0,\Delta]})\, \mathrm{d}q^e_\phi(\omega_{[0,\bar{t}^i]})$$
$$\tag{33}$$

$$\geq \int \Bigg[ \log p(y^i_{\bar{t}^i+\Delta} \mid h^i_{(\bar{t}^i+\Delta)^-}, \tilde{\omega}_{[0,\Delta]}, \omega_{[0,\bar{t}^i]})$$
$$- \log\left(\frac{q^d_\phi(\tilde{\omega}_{[0,\Delta]})}{p(\tilde{\omega}_{[0,\Delta]})}\right) - \log\left(\frac{q^e_\phi(\omega_{[0,\bar{t}^i]})}{p(\omega_{[0,\bar{t}^i]})}\right)\Bigg] \mathrm{d}q^d_\phi(\tilde{\omega}_{[0,\Delta]})\, \mathrm{d}q^e_\phi(\omega_{[0,\bar{t}^i]}) \tag{34}$$

$$= \mathbb{E}_{q^e_\phi(\omega_{[0,\bar{t}^i]})}\Big\{\mathbb{E}_{q^d_\phi(\tilde{\omega}_{[0,\Delta]})}\Big[\log p(y^i_{\bar{t}^i+\Delta} \mid \tilde{\omega}_{[0,\Delta]}, \omega_{[0,\bar{t}^i]}, h^i_{(\bar{t}^i+\Delta)^-})\Big]\Big\}$$
$$- D_{\mathrm{KL}}[q^d_\phi(\tilde{\omega}_{[0,\Delta]}) \parallel p(\tilde{\omega}_{[0,\Delta]})] - D_{\mathrm{KL}}[q^e_\phi(\omega_{[0,\bar{t}^i]}) \parallel p(\omega_{[0,\bar{t}^i]})], \tag{35}$$

where Eq. 31 follows from the standard pairwise independence assumption of inputs and weights, Eq. 32 is due to the independence of the OU priors, and Eq. 34 follows from Jensen's inequality.

The ELBO objective is maximized for $q^d_\phi(\tilde{\omega}_{[0,\Delta]}) = p(\tilde{\omega}_{[0,\Delta]} \mid h^i_{\bar{t}^i+\Delta}, \omega_{[0,\bar{t}^i]})$ and $q^e_\phi(\omega_{[0,\bar{t}^i]}) = p(\omega_{[0,\bar{t}^i]} \mid h^i_{\bar{t}^i+\Delta})$. We can see this by substituting in the true weight posteriors

into Eq. 34. This yields

$$\int \left[ \log p(y^i_{\bar{t}^i+\Delta} \mid h^i_{(\bar{t}^i+\Delta)^-}, \tilde{\omega}_{[0,\Delta]}, \omega_{[0,\bar{t}^i]}) - \log \left( \frac{p(\tilde{\omega}_{[0,\Delta]} \mid h^i_{\bar{t}^i+\Delta}, \omega_{[0,\bar{t}^i]})}{p(\tilde{\omega}_{[0,\Delta]})} \right) \right.$$

$$\left. - \log \left( \frac{p(\omega_{[0,\bar{t}^i]} \mid h^i_{\bar{t}^i+\Delta})}{p(\omega_{[0,\bar{t}^i]})} \right) \right] \mathrm{d}p(\tilde{\omega}_{[0,\Delta]} \mid h^i_{\bar{t}^i+\Delta}, \omega_{[0,\bar{t}^i]}) \, \mathrm{d}p(\omega_{[0,\bar{t}^i]} \mid h^i_{\bar{t}^i+\Delta}) \tag{36}$$

$$= \int \log \left( \frac{p(y^i_{\bar{t}^i+\Delta} \mid h^i_{(\bar{t}^i+\Delta)^-}, \tilde{\omega}_{[0,\Delta]}, \omega_{[0,\bar{t}^i]}) p(\tilde{\omega}_{[0,\Delta]}, \omega_{[0,\bar{t}^i]})}{p(\tilde{\omega}_{[0,\Delta]}, \omega_{[0,\bar{t}^i]} \mid h^i_{(\bar{t}^i+\Delta)^-}, y^i_{\bar{t}^i+\Delta})} \right) \mathrm{d}p(\tilde{\omega}_{[0,\Delta]}, \omega_{[0,\bar{t}^i]} \mid h^i_{\bar{t}^i+\Delta}) \tag{37}$$

$$= \int \log \left( \frac{p(y^i_{\bar{t}^i+\Delta}, \tilde{\omega}_{[0,\Delta]}, \omega_{[0,\bar{t}^i]}) \mid h^i_{(\bar{t}^i+\Delta)^-})}{p(\tilde{\omega}_{[0,\Delta]}, \omega_{[0,\bar{t}^i]} \mid h^i_{(\bar{t}^i+\Delta)^-}, y^i_{\bar{t}^i+\Delta})} \right) \mathrm{d}p(\tilde{\omega}_{[0,\Delta]}, \omega_{[0,\bar{t}^i]} \mid h^i_{\bar{t}^i+\Delta}) \tag{38}$$

$$= \int \log \left( \frac{p(\tilde{\omega}_{[0,\Delta]}, \omega_{[0,\bar{t}^i]} \mid h^i_{(\bar{t}^i+\Delta)^-}, y^i_{\bar{t}^i+\Delta}) p(y^i_{\bar{t}^i+\Delta} \mid h^i_{(\bar{t}^i+\Delta)^-})}{p(\tilde{\omega}_{[0,\Delta]}, \omega_{[0,\bar{t}^i]} \mid h^i_{(\bar{t}^i+\Delta)^-}, y^i_{\bar{t}^i+\Delta})} \right) \mathrm{d}p(\tilde{\omega}_{[0,\Delta]}, \omega_{[0,\bar{t}^i]} \mid h^i_{\bar{t}^i+\Delta})$$

$$\tag{39}$$

$$= \int \log p(y^i_{\bar{t}^i+\Delta} \mid h^i_{(\bar{t}^i+\Delta)^-}) \, \mathrm{d}p(\tilde{\omega}_{[0,\Delta]}, \omega_{[0,\bar{t}^i]} \mid h^i_{\bar{t}^i+\Delta}) \tag{40}$$

$$= \log p(y^i_{\bar{t}^i+\Delta} \mid h^i_{(\bar{t}^i+\Delta)^-}), \tag{41}$$

where Eq. 38 follows again from prior independence of input data and weights and the fact that $h^i_{\bar{t}^i+\Delta} = h^i_{(\bar{t}^i+\Delta)^-} \cup \{y^i_{\bar{t}^i+\Delta}\}$.

# G  Numerical Computation of the Evidence Lower Bound

The objective of our BNCDE is the evidence lower bound (ELBO), which is approximated after the forward pass of the numerical SDE solver.

Recall that, for an observation $i$, the ELBO is given by

$$\log p(y^i_{\bar{t}^i+\Delta} \mid h^i_{(\bar{t}^i+\Delta)-}) \geq \mathbb{E}_{q^e_\phi(\omega_{[0,\bar{t}^i]})}\Big\{ \mathbb{E}_{q^d_\phi(\tilde{\omega}_{[0,\Delta]})}\Big[ \log p(y^i_{\bar{t}^i+\Delta} \mid \tilde{\omega}_{[0,\Delta]}, \omega_{[0,\bar{t}^i]}, h^i_{(\bar{t}^i+\Delta)-}) \Big] \Big\}$$
$$- D_{\mathrm{KL}}[q^d_\phi(\tilde{\omega}_{[0,\Delta]}) \parallel p(\tilde{\omega}_{[0,\Delta]})] - D_{\mathrm{KL}}[q^e_\phi(\omega_{[0,\bar{t}^i]}) \parallel p(\omega_{[0,\bar{t}^i]})]. \quad (42)$$

We approximate the expectations with respect to $q^e_\phi(\omega_{[0,\bar{t}^i]})$ and $q^d_\phi(\omega_{[0,\Delta]})$ with Monte Carlo samples $\omega^j_{[0,\bar{t}^i]}$ and $\tilde{\omega}^k_{[0,\Delta]}$ of the weight trajectories from the numerical SDE solver. That is, we compute the expected likelihood via

$$\mathbb{E}_{q^e_\phi(\omega_{[0,\bar{t}^i]})}\Big\{ \mathbb{E}_{q^d_\phi(\tilde{\omega}_{[0,\Delta]})}\Big[ \log p(y^i_{\bar{t}^i+\Delta} \mid \tilde{\omega}_{[0,\Delta]}, \omega_{[0,\bar{t}^i]}, h^i_{(\bar{t}^i+\Delta)-}) \Big] \Big\} \quad (43)$$

$$\approx \frac{1}{J} \sum_{\omega^j_{\underline{[0,\bar{t}^i]}} \sim q^e_\phi} \Big\{ \frac{1}{K} \sum_{\tilde{\omega}^k_{\underline{[0,\Delta]}} \sim q^d_\phi} \Big[ \log p(y^i_{\bar{t}^i+\Delta} \mid \tilde{\omega}^k_{\underline{[0,\Delta]}}, \omega^j_{\underline{[0,\bar{t}^i]}}, h^i_{(\bar{t}^i+\Delta)-}) \Big] \Big\}, \quad (44)$$

where $\underline{[0,\bar{t}^i]}$ and $\underline{[0,\Delta]}$ are grid approximations of $[0,\bar{t}^i]$ and $[0,\Delta]$, respectively.

The Kullback-Leibler (KL) divergences between priors and variational posteriors on path space can also be computed through Monte Carlo approximations. Here, we focus on the KL-divergence in the encoder component; the decoder part follows analogously. Recall that we seek to compute the KL-divergence between the solutions of the SDEs

$$\mathrm{d}\omega_t = \begin{cases} g^e_\phi(\omega_t)\,\mathrm{d}t + \sigma\,\mathrm{d}B_t \\ h^e(\omega_t)\,\mathrm{d}t + \sigma\,\mathrm{d}B_t. \end{cases} \quad (45)$$

Previously, Tzen & Raginsky (2019) and Li et al. (2020) have shown that for two stochastic differential equations with shared diffusion coefficient $\sigma$, the KL-divergence on path space satisfies

$$D_{\mathrm{KL}}[q^e_\phi(\omega_{[0,\bar{t}^i]}) \parallel p(\omega_{[0,\bar{t}^i]})] = \mathbb{E}_{q^e_\phi(\omega_{[0,\bar{t}^i]})}\left[ \int_0^{\bar{t}^i} \big((g^e_\phi(\omega_t) - h^e(\omega_t))/\sigma\big)^2 \,\mathrm{d}t \right]. \quad (46)$$

Therefore, it can be approximated with Monte Carlo samples from the numerical SDE solver. That is, we compute

$$\mathbb{E}_{q^e_\phi(\omega_{[0,\bar{t}^i]})}\left[ \int_0^{\bar{t}^i} \big((g^e_\phi(\omega_t) - h^e(\omega_t))/\sigma\big)^2 \,\mathrm{d}t \right] \approx \frac{1}{J} \sum_{\omega^j_{\underline{[0,\bar{t}^i]}} \sim q^e_\phi} \left[ \int_{\underline{[0,\bar{t}^i]}} \big((g^e_\phi(\omega^j_t) - h^e(\omega^j_t))/\sigma\big)^2 \,\mathrm{d}t \right],$$
$$(47)$$

where $\int_{\underline{[0,\bar{t}^i]}}(\cdot)\,\mathrm{d}t$ denotes the integral approximation from the SDE solver. In other words, for each Monte Carlo particle $j$, we pass the KL-divergence term as additional state through the SDE solver and then take the Monte Carlo average at time $\bar{t}^i$.

# H    IMPLEMENTATION DETAILS

In the following, we summarize the implementation details of our BNCDE and TE-CDE. Experiments were carried out on $1\times$ NVIDIA A100-PCIE-40GB. Training of BNCDE and TE-CDE took approximately 34 hours and 22 hours, respectively.

The hyperparameters of TE-CDE were taken from the original work (Seedat et al., 2022). For both training and testing, the dropout probabilities in the prediction head were set to $p = 0.1$. As in our BNCDE, we included the time covariate $t$ as an input to TE-CDE, as this improved performance.

We optimized the hidden layers of the neural SDEs drift over $\{16, (16, 16), (16, 64, 16),$ $(16, 64, 64, 16), (16, 64, 64, 64, 16)\}$ and the diffusion coefficient $\sigma$ over $\{0.01, 0.001, 0.0001\}$. We used the same neural SDE configuration for both the encoder and the decoder. Importantly, we noticed that higher diffusions generally lead to more conservative estimates of the posterior predictive distribution, which may be desirable in medical practice. Our performance metric was the ELBO objective on the validation set. We only optimized the hyperparameters for the prediction window $\Delta = 1$ and then used the same configurations for $\Delta = 2, \ldots, 5$. For better comparability, our BNCDE had the same hyperparameter specifications as TE-CDE where possible.

For both our BNCDE and TE-CDE, we increased the learning rates of the linear embedding and prediction networks compared to the original TE-CDE implementation for more efficient training (Kidger et al., 2020). Further, we used cubic Hermite splines with backward differences in the neural CDEs, which have been shown to be superior to linear interpolation (Morrill et al., 2021). As we want to understand the true source of performance gain, we removed the balancing loss term in TE-CDE for comparability. However, we emphasize that the balancing loss term could easily be added to our BNCDE, as we show in our extended analysis in Supplement N).

Table 3 lists the selected hyperparameters.

| Component | Hyperparameter | BNCDE (ours) | TE-CDE (Seedat et al., 2022) |
|---|---|---|---|
| General | Batch size | 64 | 64 |
| | Optimizer | Adam (Kingma & Ba, 2015) | Adam |
| | Max. number of epochs | 500 | 500 |
| | Patience | 10 | 10 |
| | MC samples training | 10 | 10 |
| | MC samples prediction | 100 | 100 |
| | Hidden state size $d_z$ | 8 | 8 |
| | Interpolation method | Cubic Hermite splines | Cubic Hermite splines |
| Differential equation solver | Solver | Euler-Maruyama | Euler |
| | Step size | Adaptive | Adaptive |
| Embedding network | Learning rate | $10^{-3}$ | $10^{-3}$ |
| | Hidden layers | — | — |
| | Output activation | — | — |
| Neural CDEs | Learning rate | — | $10^{-4}$ |
| | Hidden layers | 2 | 2 |
| | Hidden dimensions | (128, 128) | (128, 128) |
| | Hidden activations | ReLU | ReLU |
| | Output activation | tanh | tanh |
| Neural SDEs | Learning rate | $10^{-4}$ | — |
| | Hidden layers | 5 | — |
| | Hidden dimensions | (16, 64, 64, 64, 16) | — |
| | Hidden activations | ReLU | — |
| | Output activation | — | — |
| | Diffusion coefficient | 0.001 | — |
| Prediction network | Learning rate | $10^{-3}$ | $10^{-3}$ |
| | Hidden layers | — | — |
| | Output activation | ( — , softplus) | — |
| | Dropout probability | — | 0.1 |
| Intensity network (only for Supplement K) | Learning rate | $10^{-3}$ | $10^{-3}$ |
| | Hidden layers | — | — |
| | Output activation | sigmoid | sigmoid |
| Balancing network (only for Supplement L) | Learning rate | $10^{-3}$ | $10^{-3}$ |
| | Hidden layers | — | — |
| | Output activation | — | — |
| | Balancing hyperparameter | 0.01 | 0.01 |

Table 3: Hyperparameters of BNCDE and TE-CDE. Hyperparameters of TE-CDE were taken from (Seedat et al., 2022). Our BNCDE used the same hyperparameters where possible.

# I EVALUATION OF OUTCOME UNCERTAINTY

We now assess the quality of the estimated outcome (*aleatoric*) uncertainty. In our BNCDE, outcome uncertainty is captured in the likelihood variance $\Sigma_{\bar{T}+\Delta}$ via

$$Y_{\bar{T}+\Delta}[a'_{(\bar{T},\bar{T}+\Delta]}] \mid (\tilde{\omega}_{[0,\Delta]}, \omega_{[0,\bar{T}]}, H_{\bar{T}}) \sim \mathcal{N}(\mu_{\bar{T}+\Delta}, \Sigma_{\bar{T}+\Delta}). \tag{48}$$

Outcome uncertainty is due to the randomness that is inherent to the data-generating process. It is irreducible in the sense that, even if we had infinite training data and hence zero model uncertainty, the potential outcome would still be random.

For a given patient history $H_{\bar{T}}$ and future sequence of treatments $a'_{(\bar{T},\bar{T}+\Delta]}$, computing the true outcome uncertainty is intractable. Instead, we can check whether our BNCDE produces informative estimates of the outcome uncertainty. Variational Bayesian methods may heavily over- or underestimate the outcome uncertainty, yielding uninformative estimates. Hence, through our analysis, we provide insights that our BNCDE produces meaningful estimates of outcome uncertainty for different patient histories and future sequences of treatments. In particular, we show that there is a direct relation between estimated outcome uncertainty and prediction error. Intuitively, prediction errors should be larger for outcomes with higher outcome uncertainty. This means that if the potential outcome for a particular patient history and future sequence of treatments is subject to a large variance, the prediction errors should increase accordingly.

The results in Fig. 6 show the relation between prediction errors and estimated outcome uncertainty for varying prediction windows. For each patient, we compute the Monte Carlo average of the estimated outcome uncertainty $\Sigma_{\bar{T}+\Delta}$ as well as the average point prediction error. We notice that our BNCDE provides varying levels of outcome uncertainty for different patients and future sequences of treatments. Further, prediction errors are highly *correlated* with estimated outcome uncertainties. Therefore, our BNCDE generates meaningful estimated outcome uncertainties. Importantly, this is another key advantage of our BNCDE over approaches such as TE-CDE (Seedat et al., 2022) with MC dropout (Gal & Ghahramani, 2016), which are not capable of estimating potential outcome uncertainties.

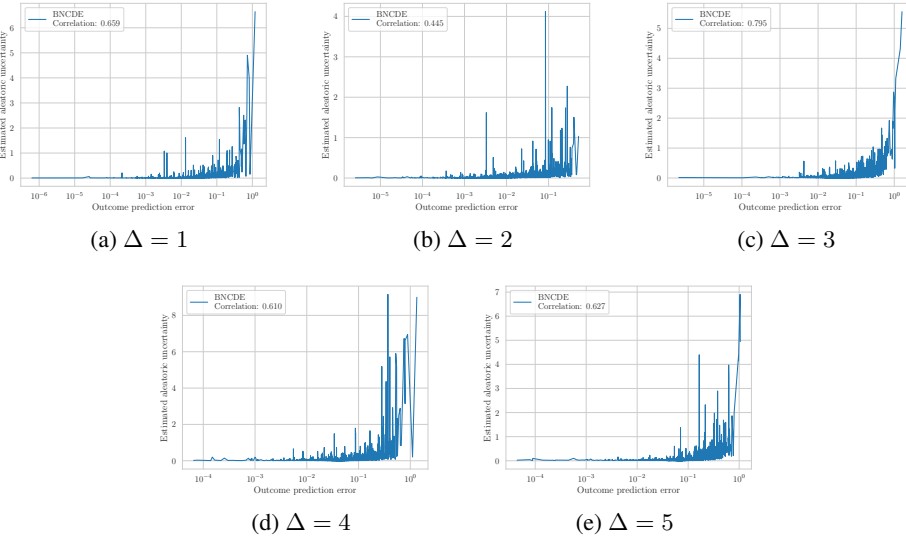

(a) $\Delta = 1$     (b) $\Delta = 2$     (c) $\Delta = 3$

(d) $\Delta = 4$     (e) $\Delta = 5$

Figure 6: We analyze the association between estimated outcome uncertainty and prediction error. The prediction errors are highly correlated with the estimated outcome uncertainties.

## J    ADDITIONAL PREDICTION WINDOWS

We repeat the experiments from Section 5 for the prediction windows $\Delta = 4$ and $\Delta = 5$. We report the results in Figures 7 and 8, respectively. The results are consistent with our main paper: (i) The credible intervals generated by our BNCDE remain *faithful* and *sharp*. In contrast, TE-CDE produces unfaithful CrIs. (ii) The point estimates of the outcome by our BNCDE are more stable under increasing noise in the data generation. (iii) The model uncertainty of our BNCDE is better calibrated.

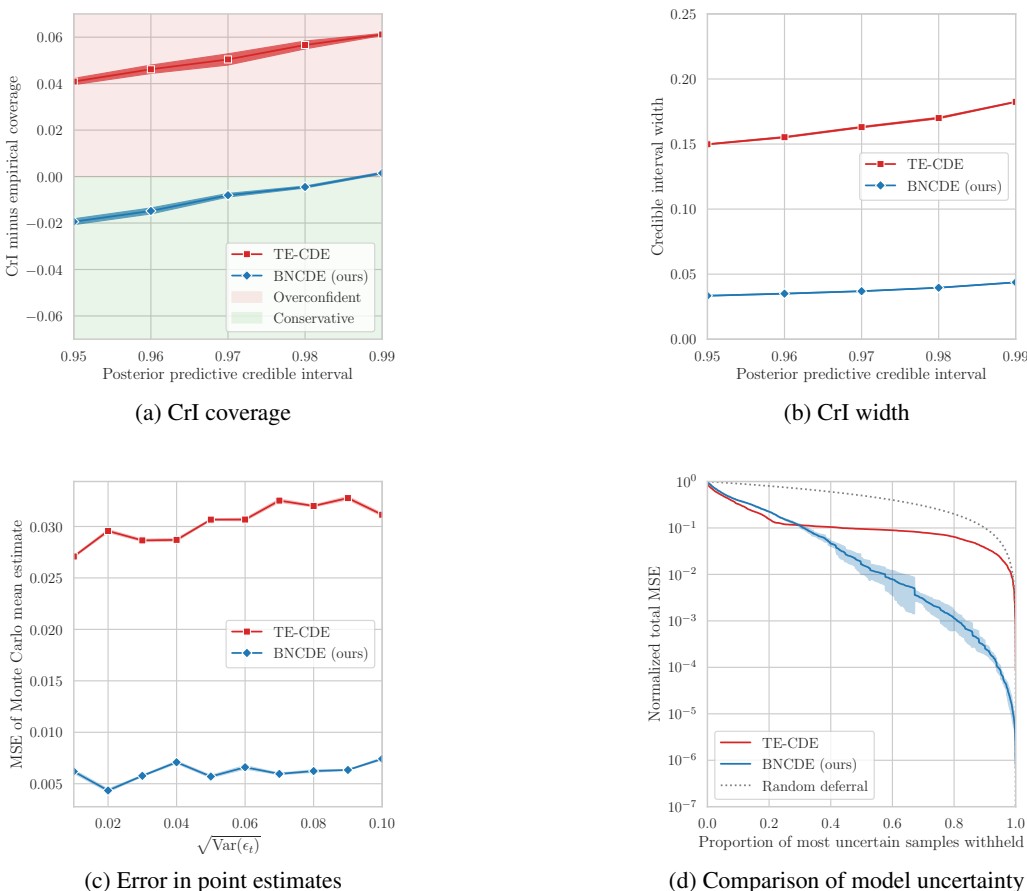

(a) CrI coverage

(b) CrI width

(c) Error in point estimates

(d) Comparison of model uncertainty

Figure 7: We repeat our experiments from Section 5. Reported are the results for the prediction window $\Delta = 4$.

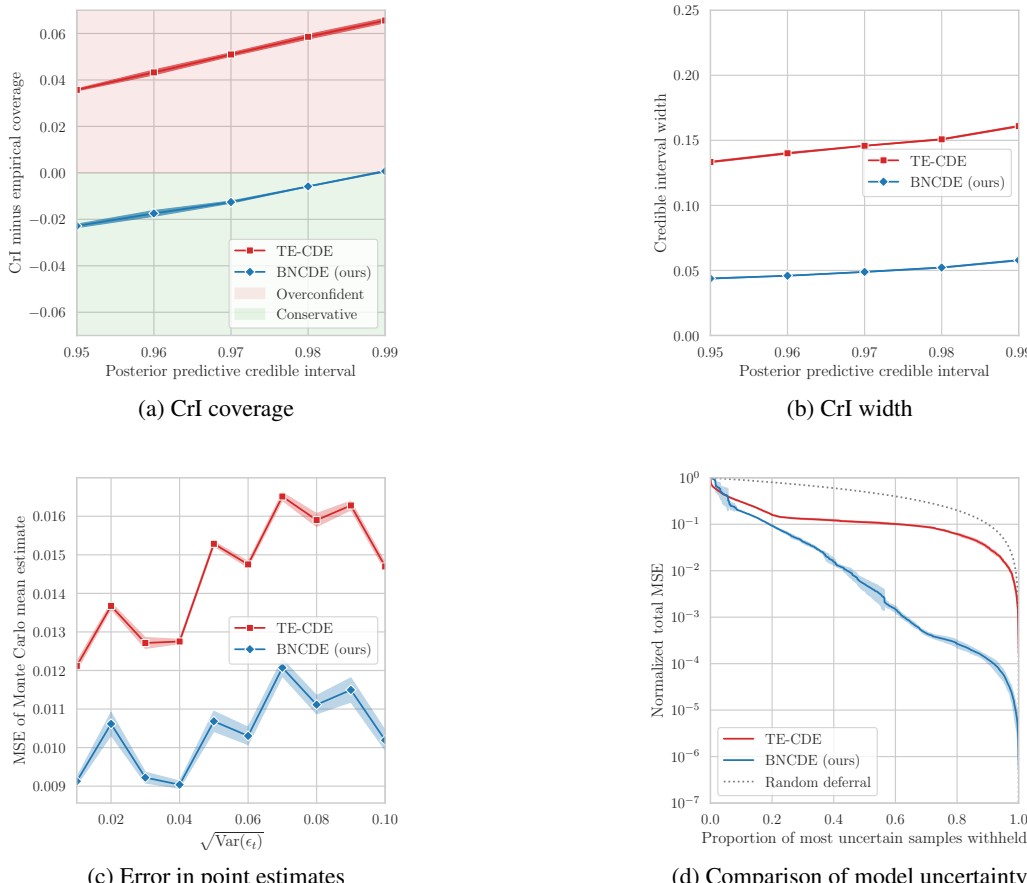

(a) CrI coverage

(b) CrI width

(c) Error in point estimates

(d) Comparison of model uncertainty

Figure 8: We repeat our experiments from Section 5. Reported are the results for the prediction window $\Delta = 5$.

# K    EXTENSION TO INFORMATIVE SAMPLING

We show that our BNCDE is superior over TE-CDE, even when used together with the informative sampling framework from Vanderschueren et al. (2023). The motivation for this is that the observation times of an outcome may themselves contain valuable information about the health status of a particular patient. For example, patients in a more severe state of health are likely to be visited more often, leading to a correspondingly higher observation intensity. To address such informative sampling, Vanderschueren et al. (2023) propose to weight the training objective with the inverse observation intensity. To achieve this, they introduce an additional prediction head for TE-CDE, which estimates the observation intensities. The inverse estimated observation intensities are then used to weight the training objective, which has been shown to improve performance in outcome estimation. This extension of TE-CDE is called the TESAR-CDE.

**Changes to tumor growth simulator:** We extend our simulation setup as follows. In the tumor growth simulator (see Supplement B), the observation times are determined by an intensity process with an intensity function

$$\zeta_t^i = \text{sigmoid}\left[\gamma\left(\frac{\bar{D}_t^i}{D} - \frac{1}{2}\right)\right], \tag{49}$$

where $\gamma$ controls the sampling informativeness, $D = 13\text{cm}$ and $\bar{D}_t^i$ is the average tumor diameter over the last 15 days. For higher values of $\gamma$, the total number of observed outcomes reduces but the informativeness of each observation timestamp increases. We later change $\gamma = 2.0$ to introduce informative sampling.

**Changes to our BNCDE:** We can easily incorporate the inverse intensity weighting into our BNCDE to account for informative sampling. The final hidden representation $\tilde{Z}_\Delta^i$ of patient $i$ is additionally passed through a second prediction head $\eta_\phi^\zeta$ to estimate the observation intensity via

$$\hat{\zeta}_{\bar{t}^i+\Delta}^i = \eta_\phi^\zeta(\tilde{Z}_\Delta^i). \tag{50}$$

The evidence lower bound (ELBO) for a patient observation $i$ is then weighted with the inverse estimated observation intensity, i.e.,

$$\text{ELBO}^i/\hat{\zeta}_{\bar{t}^i+\Delta}^i. \tag{51}$$

**Experiments:** We repeat the experiments from Section 5, where we increase the level of informativeness $\gamma$ from 1.0 to 2.0. We focus on the prediction window $\Delta = 1$ with inverse intensity weighting enabled for both methods.

We benchmark our BNCDE with TESAR-CDE. For TESAR-CDE, we enable MC dropout in the outcome prediction head $\eta_\phi^p$ but not in the intensity prediction head $\eta_\phi^\zeta$. We provide implementation details of $\eta_\phi^\zeta$ in Supplement H. For both TESAR-CDE and our extended BNCDE, optimization of $\eta_\phi^\zeta$ is performed as in the original work by Vanderschueren et al. (2023). That is, the error in the estimated observation intensity is not propagated through the whole computational graph but only through $\eta_\phi^\zeta$.

**Results:** Fig 9 shows, that in the informative sampling setting, our method remains superior. Our BNCDE (i) produces more faithful and sharper credible interval approximations; (ii) generates more stable point estimates of the outcome under increasing noise; and (iii) has a better calibrated model uncertainty. In sum, this demonstrates the effectiveness of our BNCDE even for informative sampling.

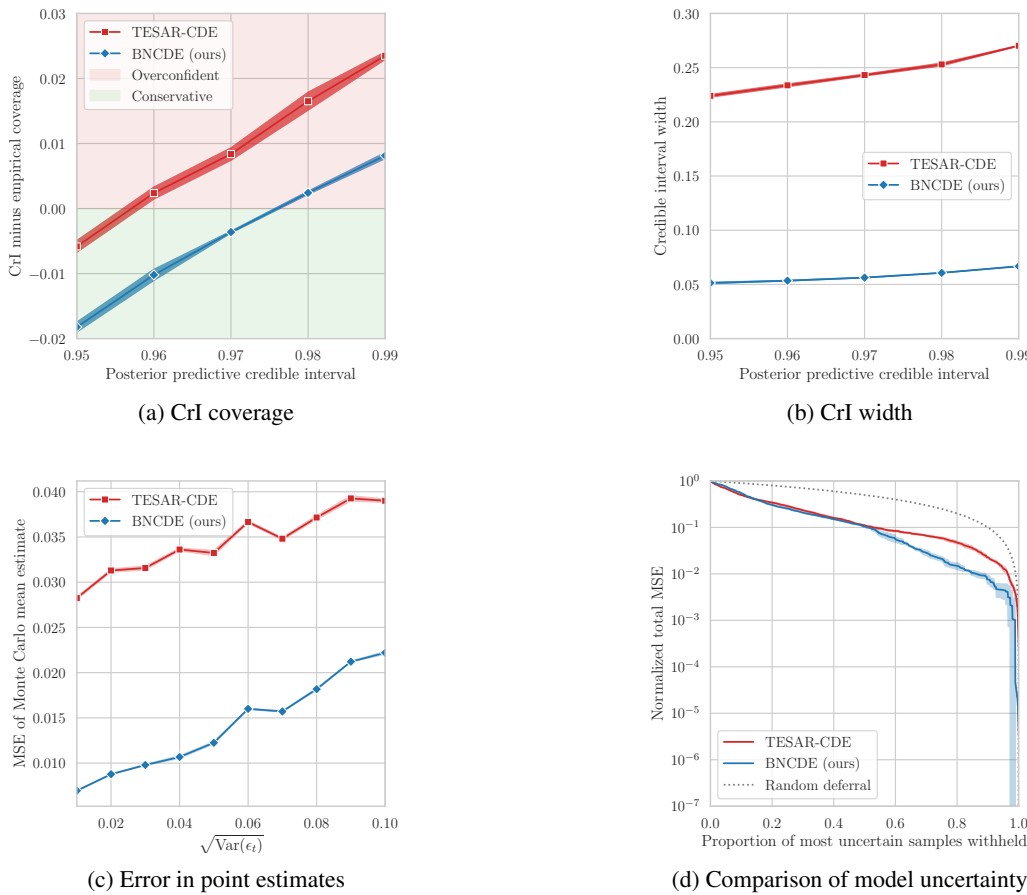

(a) CrI coverage

(b) CrI width

(c) Error in point estimates

(d) Comparison of model uncertainty

Figure 9: We repeat the experiments for settings with informative sampling. For this, we increase the informativeness parameter in the tumor growth simulator to $\gamma = 2.0$. Following (Vanderschueren et al., 2023), we weight the training objective with the estimated inverse observation intensity. We benchmark our extended BNCDE against TESAR-CDE (Vanderschueren et al., 2023), for which we enable MC dropout in the outcome prediction head.

## L    REAL-WORLD AND SEMI-SYNTHETIC DATA

We provide further insights that our BNCDE remains reliable on different data sets. For this, we use the MIMIC-extract (Wang et al., 2020). In particular, we use the same preprocessing of MIMIC-III (Johnson et al., 2016) as in earlier studies (Melnychuk et al., 2022). The MIMIC-extract provides intensive care unit data at irregularly recorded times measured in units of hours.

We also examine whether our BNCDE is compatible with balanced representations. While it is often used in the literature for mitigating confounding bias (Bica et al., 2020; Melnychuk et al., 2022; Seedat et al., 2022), it actually reduces estimation variance (Shalit et al., 2017), e.g., due to overlap violations, which is particularly relevant in the time-varying setting.

### L.1    EXPERIMENTAL SETUP

**Real-world data:** We consider patients with 30 observation times. For each patient, we select time-varying patient covariates *heart rate*, *red blood cell count*, *sodium, mean blood pressure*, *systemic vascular resistance*, *glucose*, *chloride urine*, *glascow coma scale total*, and *hematocrit and positive end-expiratory pressure set*. Additionally, we use *gender* and *age* as static covariates. We have additional access to discrete treatments *vasopressors* and *mechanical ventilation*. We seek to predict the outcome *diastolic blood pressure*. The remaining covariates serve as unobserved confounders.

We use a 60/15/25 split for training, validation, and testing on $N = 10,000$ observations.

**Semi-synthetic data generation:** In order to generate treatment assignments and outcomes, we proceed as follows:

(i) We only consider patient histories with 30 to 50 observation times. For each patient, we then select time-varying covariates (1) *heart rate*, (2) *red blood cell count*, (3) *sodium*, (4) *mean blood pressure* and (5) *systemic vascular resistance*, and static covariates (6) *gender* and (7) *age*. Importantly, these covariates are observed *irregularly* at different hourly levels. Let $K = 7$ be the number of covariates. We write $X^i_{t_j}(k)$ when referring to covariate $k \in \{1, 2, \ldots, 7\}$ of patient $i$ at hour $j$.

(ii) We forward-fill each patient covariate in time. Note that this is done only for data generation, not for training and testing, as our model is designed to handle irregularly sampled data.

(iii) Then, we simulate outcomes of patient $i$ for a burn-in period of $H = 0, \ldots, 4$ hours via

$$y^i_{t_{H+1}} = -\sum_{h=0}^{H} \frac{1}{h+1} \tanh\left(\frac{1}{2}\sum_{k=1}^{2} \beta_k X^i_{t_{H-h}}(k)\right) y^i_{t_h} + \epsilon^i_{t_h}(y), \tag{52}$$

where $y^i_{t_0} = 1$ for all $i$,, $\beta_k \sim \mathcal{N}(0, 1)$ are randomly sampled covariate coefficients, and $\epsilon^i_{t_h}(y) \sim \mathcal{N}(0, 0.01^2)$ are i.i.d. outcome noise terms.

(iv) Next, we sample the treatment assignments and patient outcomes at hours $j > 5$. At each time $t_j$, the binary treatment mechanism depends on the patient history in order to introduce confounding. Specifically, the treatment probabilities are parameterized by

$$p^i_{t_j}(a) = \text{sigmoid}\left(\tanh\left(\frac{1}{K}\sum_{k=1}^{K} \beta_k X^i_{t_{j-1}}(k)\right) + \tanh\left(y^i_{t_{j-1}}\right) + \epsilon^i_{t_j}(a)\right), \tag{53}$$

where $\epsilon^i_{t_j}(a) \sim \mathcal{N}(0, 0.03^2)$ are i.i.d. noise terms. Treatment assignments are then sampled via

$$a^i_{t_j} \sim \text{Ber}(p^i_{t_j}(a)). \tag{54}$$

We simulate outcomes via

$$y^i_{t_{j+1}} = -\sum_{h=0}^{4} \frac{1}{h+1} \tanh\left(\frac{1}{2}\sum_{k=1}^{2} \beta_k X^i_{t_{j-h}}(k)\right) y^i_{t_j} + \cos\left(y^i_{t_j} a^i_{t_j}\right) + \epsilon^i_{t_{j+1}}(y). \tag{55}$$

(v) Since our BNCDE is designed for irregularly sampled data, we do *not* use the forward-filled covariates but the *original, irregularly sampled* covariates for training and testing. Further, we

apply an observation mask to the outcomes with observation probabilities

$$\text{sigmoid}\left(y_{t_j}^i\right) \tag{56}$$

Further, we mask all outcomes for which no covariates are observed in the future.

We use a 60/15/25 split for training, validation and testing on $N = 32,673$ observations.

## L.2 BNCDE with Balanced Representations

For both the real-world data and the semi-synthetic data, our BNCDE is trained with balanced representations (Bica et al., 2020; Melnychuk et al., 2022; Seedat et al., 2022) in order to reduce finite-sample estimation variance. We provide a discussion on balanced representations in Supplement N. To this end, we follow the implementation as in (Seedat et al., 2022). That is, we add a second prediction head $\eta_\phi^a$ as in (Seedat et al., 2022) to make the hidden representations $Z_t$ and $\tilde{Z}_\tau$ non-predictive of the administered treatments. The treatment prediction head $\eta_\phi^a$ estimates the probability of a future treatment $p_{\tau_j}^i(a)$ at timestamp $\tau_j$. For a sequence of treatments with treatment decisions at irregular timestamps $\{\tau_1, \ldots, \tau_m\}$, the treatment prediction head is trained to *maximize* the binary cross-entropy given by

$$\text{BCE} = -\frac{1}{m}\sum_{j=1}^m a_{\tau_j}\log(\hat{p}_{\tau_j}^i(a)) + (1 - a_{\tau_j})\log(1 - \hat{p}_{\tau_j}^i(a)). \tag{57}$$

The overall objective is then to maximize the weighted sum of evidence lower bound and binary cross entropy. That is, we maximize

$$\text{ELBO} + \alpha\,\text{BCE}, \tag{58}$$

where $\alpha$ is a hyperparameter.

We choose $\alpha = 0.01$ and a single linear transformation for $\eta_\phi^a$. As proposed in the original TE-CDE paper (Seedat et al., 2022), we also use balanced representations for our baseline, TE-CDE with MC dropout. Importantly, we only use MC dropout in the outcome prediction head, not in the treatment prediction head. For both our BNCDE and TE-CDE, all hyperparameters are kept as in the main paper (see Supplement H).

## L.3 Results

For both datasets, we repeat our main experiments on the reliability of the posterior predictive distributions for $\Delta = 1$ hour ahead prediction.

**(i) Real-world data:** Fig. 10 shows the empirical coverage for the posterior predictive credible intervals along with the width. Here, our BNCDE tends to have wider intervals. However, we clearly see that estimates are much more reliable. Importantly, the credible intervals from our BNCDE coincide very accurately with the frequentist outcome quantiles. On the other hand, TE-CDE with MC dropout completely **fails** to obtain reliable credible intervals. In fact, the CrIs from TE-CDE with MC dropout out are way too overconfident (i.e., too narrow), which could lead to harmful decisions. In sum, our BNCDE therefore strongly outperforms the baseline on real-world data with balancing.

**(ii) Semi-synthetic data:** Fig. 11 shows the empirical coverage for the posterior predictive credible intervals along with the CrI width. Further, we increase the variance in the outcome noise $\epsilon_y(t)$ from 0.01 to 0.1 and repeat our robustness studies. Both methods generate estimates that are conservative and have sufficient predictive coverage. In particular, all of the reported posterior predictive credible intervals of our BNCDE contain **all** of the outcomes in the test set. We noticed that using balancing on this semi-synthetic dataset increased the conservatism of our estimates. This is desirable and shows in particular why our BNCDE is *compatible* with balancing. Further, compared with the TE-CDE baseline with MC dropout, our BNCDE has much sharper credible intervals, while providing more coverage than the baseline. Finally, we see that the point estimates of our BNCDE are again much more resistant to increasing noise in the outcome distribution. In sum, this demonstrates the effectiveness of our proposed BNCDE.

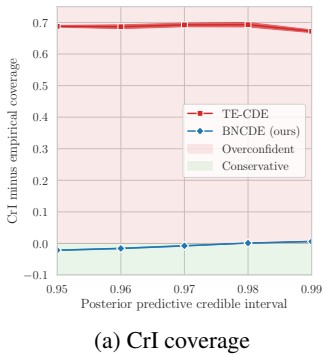
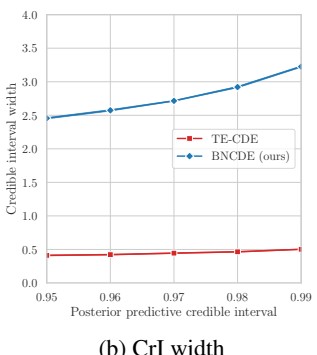

(a) CrI coverage        (b) CrI width

Figure 10: **Real-world data:** We repeat our main analysis of the reliability of the posterior predictive credible intervals with balanced representations. We benchmark our extended BNCDE against TE-CDE with MC dropout in the outcome prediction head. We include balanced representations as in the original work (Seedat et al., 2022). We report the mean and standard deviation over five different prediction runs.

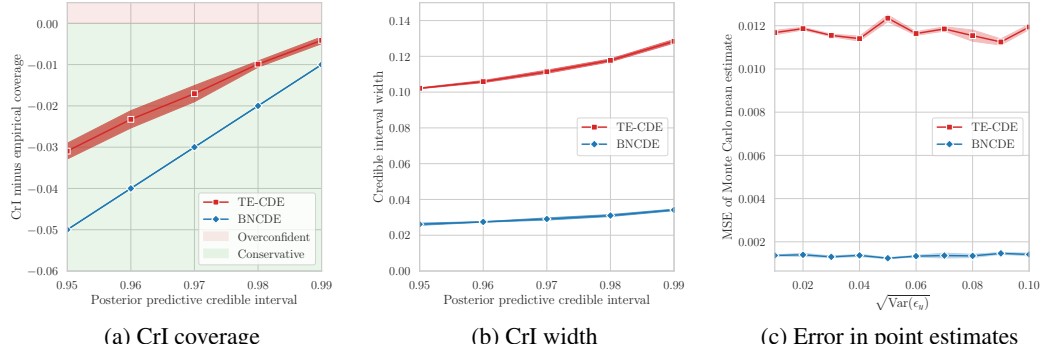

(a) CrI coverage      (b) CrI width      (c) Error in point estimates

Figure 11: **Semi-synthetic data:** We repeat our main analysis of the reliability of the posterior predictive credible intervals with balanced representations. Further, we increase the variance of the outcome noise in Eq. 55 from $0.01^2$ to $0.1^2$ in the test data. We benchmark our extended BNCDE against TE-CDE with MC dropout in the outcome prediction head. We include balanced representations as in the original work (Seedat et al., 2022). We report the mean and standard deviation over five different prediction runs.

# M  RUNTIME

Estimating treatment effects over time from electronic health records is notoriously difficult and may require complex neural architectures. Importantly, this is a common issue in the existing baselines as well as in our method. Below, we provide a discussion of why we expect that our runtime is still reasonable in practical applications.

Arguably, the computational bottleneck in training our BNCDE is solving the neural SDEs. If the neural CDE networks are very large, a high dimensional SDE has to be solved in the forward pass of the training. To better understand this, we examine empirically how the training time of our method scales with the dimension of the SDE. Thereby, we validate that training time only scales *linearly* with the dimension of the neural CDE.

For this, we train our BNCDE for a single epoch on the cancer simulation data. As detailed in Supplement B, there are $N = 10,000$ training observations and we train on batches of size $64$. All experiments were carried out on $1\times$ NVIDIA A100-PCIE-40GB.

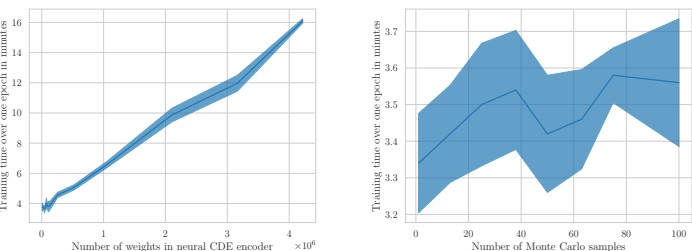

(a) We increase the number of hidden layers in the encoder neural CDE. Runtime increases approximately linearly.

(b) We increase the number of Monte Carlo samples in the SDE solver. Multiple samples can be used at little cost.

Figure 12: Runtime evaluation of our BNCDE.

First, we use the same parameterization as provided in Supplement H but change the number of hidden layers in the *neural CDE of the encoder*. That is, we increase the number of hidden layers of size 128 in the neural CDE and, hence, the dimension of the latent neural SDE. Fig. 12(left) reports the average training time in minutes along with the standard deviation over three seeds against the number of weights in neural CDE of the encoder. The training time only increases linearly with the number of weights, which mimics the usual computational complexity for any fully connected neural network.

Second, we investigate how training time scales with the number of Monte Carlo samples of the weight trajectories $\omega_{[0,T]}$ and $\tilde{\omega}_{[0,\Delta]}$. Both the approximations of (i) the expected likelihood and (ii) the Kullback-Leibler divergence in the evidence lower bound rely on Monte Carlo samples of $\omega_{[0,T]}$ and $\tilde{\omega}_{[0,\Delta]}$ (see Supplement G). Hence, training our BNCDE stabilizes for a higher number of Monte Carlo samples of $\omega_{[0,T]}$ and $\tilde{\omega}_{[0,\Delta]}$. In order to gain insight how training time of our BNCDE scales under *multiple* draws Monte Carlo draws of $\omega_{[0,T]}$ and $\tilde{\omega}_{[0,\Delta]}$ at once, we use the same hyperparameters as in our main experiments (see Supplement H), but we increase the number of Monte Carlo draws from 1 to 100. Fig. 12 (right) reports the average training time in minutes along with the standard deviation over three seeds against the number of Monte Carlo draws for both $\omega_{[0,T]}$ and $\tilde{\omega}_{[0,\Delta]}$. Training time barely increases, which shows that Monte Carlo variance in the training can be reduced at little cost.

As a side note, we emphasize that neither increasing the dimension of the latent neural SDEs nor the number of Monte Carlo samples affects the discretization error of the SDE solver. In our experiments, we found that adaptive solvers are suitable, e.g., the adaptive Euler-Maruyama scheme. Following (Chen et al., 2018), we argue that discretization errors are in practice not an issue, as modern solvers provide guarantees on accuracy.

In sum, we conclude that there are no drawbacks in using very deep neural CDE networks that are specific to our BNCDE. Still, as is the case with other approximate Bayesian methods, there are

no stability guarantees when training our method, that is, for learning very high-dimensional neural SDEs (e.g., in the billions). Rather, our work focuses on Bayesian inference for neural CDEs of moderate size as encountered in medical practice.

# N   DISCUSSION ON BALANCED REPRESENTATIONS

Some prior works on estimating heterogeneous treatment effects propose to learn balanced representations that are non-predictive of the treatment (Bica et al., 2020; Melnychuk et al., 2022; Seedat et al., 2022). The idea is to mimic randomized clinical trials and reduce finite-sample error due to estimation variance (Shalit et al., 2017). However, for time-varying treatment effects, this approach has several drawbacks:

1. Learning guarantees only exist in static (i.e., non-time-varying) settings (Shalit et al., 2017). Thus, balanced representations do not help with reducing bias due to identifiability issues induced by time-varying confounders. For this, proper adjustment methods such as G-computation or inverse-propensity weighting would be necessary (Pearl, 2009; Robins & Hernán, 2009).

2. Even in static settings, methods based on balanced representations impose invertibility assumptions on the learned representations (Shalit et al., 2017). This is highly unrealistic in time-varying settings, where representations not only incorporate information about the confounders but also about the full patient history.

3. Because invertibility is difficult to ensure, balanced representations may actually increase bias. We refer to Curth & van der Schaar (2021) and Melnychuk et al. (2024) for a detailed discussion on this issue. Again, this is particularly detrimental in time-varying settings.

In our main paper, we therefore decided not to incorporate balanced representations due to the reasons described above. This is consistent with previous works in the literature (Curth & van der Schaar, 2021; Vanderschueren et al., 2023). Nevertheless, we emphasize that balanced representations can be easily integrated into our BNCDE, see Supplement L.

In summary, balanced representations are a heuristic approach for reducing finite-sample variance. Importantly, in order to avoid confounding bias, proper adjustments such as G-computation or inverse-propensity weighting are needed.

