# OpenReview forum: "Bayesian Neural Controlled Differential Equations for Treatment Effect Estimation"
_ICLR.cc/2024/Conference — ICLR 2024 poster_

### Official Review · Reviewer_kEvk · 2023-10-27

**Soundness:** 3 good
**Presentation:** 3 good
**Contribution:** 2 fair
**Rating:** 6
**Confidence:** 3

**Summary:**

This paper presents a Bayesian Controlled Neural Differential Equations (BNCDE) framework aimed at estimating treatment effects in time-series data within a continuous time. The approach involves employing a Bayesian Neural Network to model the drift function of the CDE, with the posterior distribution of the neural network weights defined by the solution of a stochastic differential equation.
The authors outline a training regimen for the framework by optimizating the Evidence Lower Bound (ELBO) in an end-to-end manner.
On the empirical side, the author includes experiments conducted on synthetic tumor growth data, with a focus on evaluating the uncertainty estimation quality of the proposed model. The results from these experiments are compared to non-Bayesian counterparts, highlighting the performance differences between the two approaches.

**Strengths:**

This paper proposes a treatment effect estimation framework for time series in continuous time, demonstrating a better uncertainty estimation compared to the non-Bayesian counterpart with MC dropout. I briefly checked the derivation, which seems to be correct. The presentation of method is clear in most places but there are still some ambiguities that requires further clarification.

**Weaknesses:**

There are two aspects that warrant further discussion, particularly in terms of its novelty and scalability. With respect to novelty, it appears that the main formulations of Stochastic Differential Equations (SDE) and Controlled Differential Equations (CDE) are derived from prior works, and the variational inference methods employed are consistent with those utilized in existing latent SDE literature [1]. This gives an impression that the current work may be an amalgamation of these previous methodologies, adapted to extend the CDE framework. Furthermore, there are noticeable similarities between this work and [2]. Despite the authors’ footnote indicating a substantial divergence from [2], there is a potential for their framework to be adaptable for time series treatment and to incorporate CDE elements. A more comprehensive discussion of the distinctions between the current work and [2] could enhance the clarity on this matter.

In terms of scalability, the neural network weights are modeled through an SDE with a neural network drift function. This configuration implies that the input dimensionality of the SDE drift scales with the dimensionality of the weights, which could potentially escalate to millions. This raises pertinent questions regarding the computational cost and the stability of the training process under such high-dimensional circumstances. An exploration of these aspects would contribute to a more thorough understanding of the framework’s practical applicability and limitations.

[1] Li, Xuechen, et al. "Scalable gradients for stochastic differential equations." International Conference on Artificial Intelligence and Statistics. PMLR, 2020.
[2] De Brouwer, Edward, Javier Gonzalez, and Stephanie Hyland. "Predicting the impact of treatments over time with uncertainty aware neural differential equations." International Conference on Artificial Intelligence and Statistics. PMLR, 2022.

**Questions:**

1. In section 3.3, the author briefly talked about the 3 assumptions required for identifiability. To make it even more clear, the author can consider adding explicit reference or discussions regarding why those 3 assumptions can lead to identifiability.

2. Since the potential outcome is modelled through the latent embeddings, does this affect the identifiability?

3. The prediction also estimate the outcome uncertainty. But typically, this can potential lead to the a naive model with large uncertainty. I wonder the quantification quality for the aleatoric uncertainty.

---

> ### Author Response · Authors · 2023-11-19
> **Response to kEvk (i)**
>
> Thank you very much for the constructive evaluation of our paper. We are happy to address your questions below.
>
> **# Response to “Weaknesses”**
>
> **Response to (1a) neural SDEs and neural CDEs:**
>
> Thank you for giving us the opportunity to clarify the novelty of our method and how we add to the existing literature. In particular, our work makes non-trivial contributions along **three literature streams**.
>
>
> 1. We contribute to the literature on neural differential equations. Neural CDE [1] and neural SDE [2] have been proposed previously in isolation, while ours is the **first** method to combine them into a single method. Further, we are not aware of any prior work that has coupled neural SDEs in an encoder-decoder framework. More importantly, to estimate treatment effects, a simple ‘plug-n-play’ approach combining both is typically **not** sufficient. In contrast, our method extends them in a non-trivial way with additional techniques from the causal machine learning literature (e.g., informative sampling in **Supplement L**, balanced representations in **new Supplement M**). Moreover, there does not exist a single tailored Bayesian neural CDE method.
> 2. We contribute to the literature on treatment effect estimation. Existing methods for treatment effect estimation in continuous time (TE-CDE [3]) are limited to point estimates. So far, no method for uncertainty quantification in this setting existed. Ours is the **first** method that allows for rigorous, Bayesian uncertainty quantification.
> 3. We contribute to the medical literature. We show that the MC dropout [4] (as used in our baselines [3]) gives _incorrect_ uncertainty estimates, which may lead to _harmful_ decisions in medical practice. As a remedy, we offer a new method to support reliable medical decision-making over time by providing **reliable** uncertainty estimates. As such, we study an important and underexplored problem in medicine.
>
> **Action:** We carefully checked that we clearly spelled out the novelty of our work. In particular, we revised **Section 1** and emphasized our contribution to the three different literature streams.
>
> **Response to (1b) distinction from CF-ODE:**
>
> Thank you for giving us the opportunity to highlight the differences between our BNCDE and the CF-ODE [5]. The are **six important differences**:
>
>
>
> 1. CF-ODE is inherently designed to forecast the treatment effect over time of a **single**, **static** treatment. In contrast, our BNCDE is designed to estimate treatment effects for a **sequence of multiple** treatments in the future. This is a much more complex task.
> 2. Accordingly, CF-ODE builds upon a **static identifiability** framework. Here, adjusting for the measurements of the patient history $S_{t^*}(X)$ at the time of treatment, $t^*$ is assumed to be sufficient to adjust for all confounders (Assumption 2.3).
> 3. CF-ODE directly models the latent states in the decoder with a neural SDE. We do not see how this can be effectively combined with neural CDEs to model arbitrary, future sequences of multiple treatments.
> 4. All patient information in CF-ODE is contained **deterministically** in the initial state of the ODE. Hence, all **uncertainty** needs to be incorporated in a **deterministic**, initial state. We argue that this is very **inefficient**. Instead, our BNCDE captures patient information in the random variable $Z_T$. Here, all the Monte Carlo variance is propagated through the encoded trajectory $Z_{[0,T]}$, accounting for uncertainty at each point in time $t \in [0,T]$. Therefore, model uncertainty is directly incorporated into the patient encoding.
> 5. The use of a GRU in the CF-ODE encoder implies **regular sampling** of the patient trajectories. This assumption is a **crucial violation of medical reality**, where it is standard that measurements are taken at **irregular** points in time. Therefore, CF-ODE does **not** account for uncertainty due to irregular sampling.
> 6. The single treatment decision in CF-ODE is limited to **binary treatments**. Instead, our BNCDE can deal with any treatment (in our experiments, treatments are discrete).
>
> Because of the above, CF-ODE is **not** applicable to our setting.
>
> **Action:** We added a new **Supplement F** where we spell out the differences between CF-ODE and our method. In sum, CF-ODE is **not** applicable to our setting. We also refer to our new **Supplement E,** where we further highlight the limitations of neural ODE modeling.

---

> > ### Comment · Reviewer_kEvk · 2023-11-22
> >
> > Thanks for the authors' detailed response.
> > However, balanced representation and informative sampling are not new and can be adapted to previous works as well.
> >
> > I do understand that there are differences of BNCDE and CF-ODE. My question is what are the fundamental differences compared to CF-ODE such that CF-ODE cannot be adapted to incorporate such feature.
> >
> > Regarding modelling neural network weights with SDE, I am not convinced that it is scalable. For example, if a transformer with many layers, which can be billions of parameters, and you also need another network to output its weights, I doubt that it can be trained stably.

---

> ### Author Response · Authors · 2023-11-19
> **Response to kEvk (ii)**
>
> **Response to (2) scalability:**
>
> This is an excellent question. Increasing the depth of the neural CDE networks directly affects the dimension of the latent neural SDEs. Accordingly, the differential equation solver must solve a higher dimensional SDE. Dimensionality is, however, not a real problem for our method for several reasons:
>
>
>
> * (i) In **Supplement I**, we **report the average training time** of our method. Overall, the average time of our BNCDE was **comparable** to the  TE-CDE [3] baseline (approx. 34h vs 22h).
> * (ii) We provide new experiments in **Supplement N** which show that the training time increases only **linearly** with the depth of the neural CDE networks (i.e., the dimension of the neural SDE). Linear scaling is not different from any other, fully connected neural network architecture.
> * (iii) Furthermore, using multiple Monte Carlo trajectories at the same time reduces Monte Carlo variance during the training and, thus, stabilizes training. In the new **Supplement N,** we show that increasing the number of Monte Carlo samples comes at very little extra cost.
> * (iv) We did not encounter any instability issues for our method. Our results are very consistent over different seeds (see our experiments in Section 5).
> * (v) Importantly, discretization errors in SDE solvers do not increase with the dimension, but only depend on the order of the solver and the chosen time grid. Modern solvers can adaptively choose the grid size and provide error guarantees.
>
> **Action:** We added new experiments in **Supplement N**, where we show that the training time only increases linearly with the depth of the neural CDEs, i.e., dimension of the neural SDEs. Further, we demonstrate that using multiple Monte Carlo samples at the same time comes at very little extra cost.

---

> > ### Author Response · Authors · 2023-11-19
> > **Response to kEvk (iii)**
> >
> > **# Response to “Questions”**
> >
> > **Response to (1) identifiability**
> >
> > Thank you for this question. Identifiability under our assumptions has been discussed previously, and we thus happily refer to the literature (i.e., Section 2.2 in [6]). This framework is completely equivalent to [13].
> >
> > **Action:** We added an explicit reference [6] as to why our assumptions lead to identifiability.
> >
> > **Response to (2) latent embeddings**
> >
> > This is a great question. We use latent embeddings to capture non-linear dependencies in the patient data and reduce dimensionality. This is completely in line with previous works on treatment effect estimation over time [7,8,4,9] and even with works in the static setting [10,11,12].
> >
> > However, when encoding into latent representations, information on confounders in the original data may be lost and, indeed, affect identifiability. However, as we train our model to be predictive of the potential outcomes, we will not lose relevant information on confounders if the dimension of our latent representation is not too small.
> >
> > **Action:** We elaborate on the use of latent representations and the implications for representation-induced bias in **Section 4.1**. We point to an analysis between representation learning and representation-induced bias as future research, which, even in the static setting, has not yet been explored.
> >
> > **Response to (3) quality of aleatoric uncertainty**
> >
> > Thank you. We are happy to demonstrate the **quantification quality of aleatoric uncertainty** in our BNCDE (see new **Supplement J**). To this end, we performed **new experiments** where we plot estimation errors against estimates of aleatoric uncertainty, finding that our method can adequately capture uncertainty in the data-generating process.
> >
> > **Action:** We add a new experiment that demonstrates the quantification quality of aleatoric uncertainty (see new  **Supplement J**). Thereby, we show that our BNCDE provides meaningful estimates of aleatoric uncertainty.
> >
> > [1] Patrick Kidger, James Morrill, James Foster, and Terry Lyons. Neural controlled differential equations for irregular time series. In NeurIPS, 2020.
> >
> > [2] Xuechen Li, Ting-Kam Leonard Wong, Ricky T. Q. Chen, and David Duvenaud. Scalable gradients for stochastic differential equations. In AISTATS, 2020.
> >
> > [3] Nabeel Seedat, Fergus Imrie, Alexis Bellot, Zhaozhi Qian, and Mihaela van der Schaar. Continuous-time modeling of counterfactual outcomes using neural controlled differential equations. In ICML, 2022.
> >
> > [4] Yarin Gal and Zoubin Ghahramani. Dropout as a Bayesian approximation: Representing model uncertainty in deep learning. In ICML, 2016.
> >
> > [5] Edward De Brouwer, Javier Gonzalez Hernandez, and Stephanie Hyland. Predicting the impact of treatments over time with uncertainty aware neural differential equations. In AISTATS, 2022.
> >
> > [6] Andrew Ying. Causality of Functional Longitudinal Data. arXiv preprint, 2206.12525, 2022.
> >
> > [7] Ioana Bica, Ahmed M. Alaa, James Jordon, and Mihaela van der Schaar. Estimating counterfactual treatment outcomes over time through adversarially balanced representations. In ICLR, 2020.
> >
> > [8] Valentyn Melnychuk, Dennis Frauen, and Stefan Feuerriegel. Causal transformer for estimating counterfactual outcomes. In ICML, 2022.
> >
> > [9] Toon Vanderschueren, Alicia Curth, Wouter Verbeke, and Mihaela van der Schaar. Accounting for Informative sampling when learning to forecast treatment outcomes over time. In ICML, 2023.
> >
> > [10] Fredrik D. Johansson, Uri Shalit, and David Sontag. Learning representations for counterfactual inference. In International Conference on Machine Learning, 2016.
> >
> > [11] Uri Shalit, Fredrik D. Johansson, and David Sontag. Estimating individual treatment effect: Generalization bounds and algorithms. In International Conference on Machine Learning, 2017.
> >
> > [12] Fredrik D. Johansson, Uri Shalit, Nathan Kallus, and David Sontag. Generalization bounds and representation learning for estimation of potential outcomes and causal effects. Journal of Machine Learning Research, 23:7489–7538, 2022.
> >
> > [13] Helene C. Rytgaard, Thomas A. Gerds, and Mark J. van der Laan. Continuous-time targeted minimum loss-based estimation of intervention-specific mean outcomes. The Annals of Statistics, 2022.

---

> ### Author Response · Authors · 2023-11-22
> **Response to kEvk (i)**
>
> Thank you very much for your response. We appreciate your feedback and we apologize if we were not clear enough in our first response.
>
> **Response to (1): novelty of balanced representations and informative sampling**
>
> Thank you. We do not claim that balancing and informative sampling is our novelty. Rather, we would like to highlight that our proposed method is capable of incorporating existing contributions from the causal ML literature.  As such, we see the ideas of balanced representations and informative sampling orthogonal to our main contribution, which is: **_Bayesian uncertainty quantification for treatment effect estimation in continuous time_**.
>
> Importantly, the **key contribution** of our method is orthogonal to the above topics such as balancing in that our method can produce highly reliable estimates of **uncertainty** (both aleatoric and epistemic). Our method achieves all of the following in a single, neural end-to-end architecture. Specifically, our method generates
>
>
>
> 1. informative **epistemic** uncertainty quantification
> 2. and meaningful **aleatoric** uncertainty quantification of the potential outcome
> 3. for **any** kind of treatment (e.g., binary, discrete, …)
> 4. and **multiple** sequences of treatments in the future
> 5. in **continuous** time.
>
> So far, there does **not exist a single method** that achieves the above. In addition, our method is also compatible with innovations from earlier research (i.e., balanced representations and informative sampling).
>
> **Response to (2): Non-applicability of CF-ODE**
>
> Thank you for your follow-up question. We apologize if we were not clear enough. CF-ODE is a method that is designed for the **_single time intervention_** setting. This is a completely **different setting** from our **_time-varying treatment effect estimation setting_**. Hence, it is **not possible to adapt to it**.
>
> _The technical challenges are located in the different settings:_
>
> - CF-ODE builds on a **static** identifiability framework. It is therefore completely designed to this specific setting. The mathematical framework of CF-ODE, i.e. the architecture, the identifiability assumption, the causal diagram, is completely designed for one, single binary treatment.
>
> - Therefore, CF-ODE can only deal with a **single, static treatment,** and this is a choice by design (that one cannot circumvent). We emphasize that this is (1) a much easier task and (2) contradicts medical reality, where schedules of multiple treatments are planned for the future. In contrast, our method can deal with a **sequence of discrete treatments** in the future.
>
> _So, why is not possible to adapt CF-ODE from the static to the time-varying setting?_
>
> - In order to account for multiple treatments in the future, it would be necessary to incorporate, for example, a **neural CDE** into CF-ODE. Both are entirely different techniques. We provide a clear distinction between neural CDE and neural ODE in our new **Supplement E**.
>
> - However, multiple treatments via neural CDEs are **not compatible** with CF-ODE. First, the mathematical derivations all build around the **single, static treatment** (see, for example, definitions of $h(t^*)$, $u_T(t-t*)$, and the causal graph (Figure 2) in Section 2 of CF-ODE [8])**.** It would therefore be necessary to develop a completely different mathematical framework, find different derivations and develop a completely different architecture to incorporate multiple treatments.
> - Further, CF-ODE **directly** models the latent states in their architecture with a neural SDE, see Eq. (5) in Section 3 of CF-ODE [8]. Because of this, it is **not possible** to directly insert a CDE into the CF-ODE architecture. To address this in method,  we design our latent states in a way they are captured with a neural CDE. Instead of directly modeling uncertainty of the latent states, **we model uncertainty through the parameterization of the latent states**. That is, we build upon a **coupled** system of neural CDEs and neural SDEs. In particular, such a method has never been proposed before.
>
> - Further, we emphasize that there are direct **limitations of how uncertainty is captured** in the CF-ODE encoder, which is, deterministically and in discrete time. This imposes strong assumptions on medical reality.
>
> Hence, we summarize: CF-ODE proceeds by (1) **directly** modeling latent states with a neural SDE and, therefore, (2) can **not be combined with neural CDEs**. Further, CF-ODE (3) builds upon a **static framework**, which (4) **does not account for discrete** treatments, and (5) completely **hinders** **the possibility for multiple treatments in the future**. All derivations and the architecture are designed for a single, static treatment. Finally, (6) it is very **limited in capturing uncertainty** through the patient input.
>
> Therefore, CF-ODE is **not at all** compatible with our setting, and we do not see how it can be extended to our setting.

---

> ### Author Response · Authors · 2023-11-22
> **Response to kEvk (ii)**
>
> **Response to (3): Scalability**
>
> Thank you for your question. We are happy to clarify why problems that arise with these types of networks are not problematic in our scenario. Importantly, our proposed method is carefully designed for uncertainty quantification in medical settings, where uncertainty estimates are of utmost importance for decision-making.
>
> - We would highlight that uncertainty quantification is typically needed for moderate-sized (rather than large) datasets in practice. The reason is that, for moderate-sized datasets, we cannot expect to learn the true data-generating mechanisms. Instead, we expect that there is uncertainty in how medical treatments affect patient health and, to deal with this, it is imperative in medicine to make treatment choices by incorporating the underlying uncertainty [7].
>
> - We emphasize that medical datasets (for which our method is designed for) typically have a moderate number of data points in the order ten thousand data points  (e.g., [2,3,4,5,6]), making the use of heavily overparameterized neural networks with multiple billions parameters often unnecessary or even infeasible. Even more, using a network architecture with multiple billions of parameters would overfit on datasets from medicine, which would completely contradict the task of our method, which is reliable uncertainty quantification.
>
> - Of note, even transformers for healthcare applications also limit the number of parameters and do not use multiple billions of parameters, thereby avoiding the aforementioned overparameterization [9, 10]. Further, we emphasize that neural CDEs typically require fewer parameters than transformers, which have a much more complex architecture.
>
> To address your point, we also show in our simulations (see **Supplement N**) that reducing Monte Carlo variance in our training comes at basically no cost. Similar to other neural architectures, training time only increases linearly with the depth of the networks. That is, it increases with $\mathcal{O}(d_\omega)$ in the notation from our paper. Hence, there are no drawbacks during training that are specific to our method.
>
> Finally, we reported all of our results in the main section over 5 different seeds. Here, we see that our method produced highly **stable** results over all seeds.
>
> [1] Toon Vanderschueren, Alicia Curth, Wouter Verbeke, and Mihaela van der Schaar. Accounting for informative sampling when learning to forecast treatment outcomes over time. In ICML, 2023.
>
> [2] Alistair E. W. Johnson, Tom J. Pollard, Lu Shen, Li-wei H. Lehman, Mengling Feng, Mohammad Ghassemi, Benjamin Moody, Peter Szolovits, Leo Anthony Celi, and Roger G. Mark. MIMIC-III, a freely accessible critical care database. Scientific Data, 3(1):160035, 2016.
>
> [3] Tom J. Pollard, Alistair E. W. Johnson, Jesse D. Raffa, Leo A. Celi, Roger G. Mark, and Omar Badawi. The eICU Collaborative Research Database, a freely available multi-center database for critical care research. Scientific Data, 5:180178, 2018.
>
> [4] Stephanie L. Hyland, Martin Faltys, Matthias H¨user, Xinrui Lyu, Thomas Gumbsch, Cristobal Esteban, Christian Bock, Max Horn, Michael Moor, Bastian Rieck, Marc Zimmermann, Dean Bodenham, Karsten Borgwardt, Gunnar R¨atsch, and Tobias M. Merz. Early prediction of circulatory failure in the intensive care unit using machine learning. Nature medicine, 26(3):364–373, 2020.
>
> [5] Patrick J. Thoral, Jan M. Peppink, Ronald H. Driessen, Eric J. G. Sijbrands, Erwin J. O. Kompanje, Lewis Kaplan, Heatherlee Bailey, Jozef Kesecioglu, Maurizio Cecconi, Matthew Churpek, Gilles Clermont, Mihaela van der Schaar, Ari Ercole, Armand R. J. Girbes, and Paul W. G. Elbers. Sharing ICU patient data responsibly under the Society of Critical Care Medicine/European Society of Intensive Care Medicine Joint Data Science Collaboration: The Amsterdam University Medical Centers Database (AmsterdamUMCdb) example. Critical Care Medicine, 49(6):563–577, 2021.
>
> [6] Niklas Rodemund, Bernhard Wernly, Christian Jung, Crispiana Cozowicz, and Andreas Koköfer. The Salzburg Intensive Care database (SICdb): an openly available critical care dataset. Intensive care medicine, 49(6):700–702, 2023.
>
> [7] Christopher R. S. Banerji, Tapabrata Chakraborti, Chris Harbron, and Ben D. MacArthur. Clinical AI tools must convey predictive uncertainty for each individual patient. Nature medicine, 2023.
>
> [8] Edward De Brouwer, Javier Gonzalez Hernandez, and Stephanie Hyland. Predicting the impact of treatments over time with uncertainty aware neural differential equations. In AISTATS, 2022.
>
> [9] Valentyn Melnychuk, Dennis Frauen, and Stefan Feuerriegel. Causal transformer for estimating counterfactual outcomes. In ICML, 2022.
>
> [10] Emmi Antikainen, Joonas Linnosmaa, Adil Umer, Niku Oksala, Markku Eskola, Mark van Gils, Jussi Hernesniemi, and Moncef Gabbouj. Transformers for cardiac patient mortality risk prediction from heterogeneous electronic health records. Scientific Reports, 13(1):3517, 2023.

---

> > ### Comment · Reviewer_kEvk · 2023-11-23
> >
> > Thanks for the authors' response. They manage to address most of my concerns. I still believe that the current framework of modelling weights is not a scalable approach and cannot be generalized to other high dimensional data types. But this is beyond the scope of this work. I highly suggest that these should be made very clearly in the main text and an in-depth discussion of the scalability issue is needed.

---

> ### Author Response · Authors · 2023-11-23
> **Response to kEvk**
>
> Thank you very much. We understand that scalability is an important issue. Therefore, we updated our work according to your suggestions. We highlighted in **Section 4.2** that our work focuses on variational inference for neural CDEs of moderate size, and that stability is unexplored for high dimensional settings. Further, we added a word of caution in  **Supplement N**. Finally, we extended our discussion on the applicability in medical practice in **Supplement A**.

---

### Official Review · Reviewer_6Zs2 · 2023-10-31

**Soundness:** 3 good
**Presentation:** 4 excellent
**Contribution:** 3 good
**Rating:** 8
**Confidence:** 3

**Summary:**

The paper proposes an estimation of the treatment effects with uncertainty in continuous time by exploiting Bayesian neural controlled differential equations. The proposed method (BNCDE) is based on an encoder-decoder architecture and allows for estimating posterior predictive distributions of the potential outcomes. In the proposed method the time dimension is modeled through a coupled system of neural-controlled differential equations and neural stochastic differential equations.

**Strengths:**

-	The paper is clearly written and identifies the gap in the literature with clear motivation (estimating treatment effects in continuos time with uncertainty quantification). The paper is self-contained and easy to follow.

-	The implementation details are well documented in both the paper/appendix and the provided code.

**Weaknesses:**

-	Considering medical applications in mind, it is important to mention: 1) possible downsides of the proposed approach; and 2) which assumptions need to be satisfied for the method to be robust.

-	The method is only compared against other neural methods. I understand those are the most natural competitors, but this ignores a large body of literature. Are there other methods which could deal with the same setup? Or not at all?

**Questions:**

-	The conclusion states: “(4) Our BNCDE is further fairly robust against noise”. It would be nice to be more explicit about what “fairly” means.

-	Please discuss possible failure modes of the method. Considering that the aim is to apply it in the medical domain, this would be relevant for anyone trying to use it.

-	In the abstract, it states: “However, existing methods for this task are limited to point estimates of the potential outcomes, whereas uncertainty estimates have been ignored.”. I believe this is an incorrect assumption. There are methods which take uncertainty into account. Are you specifically referring here to neural methods?

-	The confounding problem is not discussed in the proposed approach. As far as I understand, the baseline method is able to deal with confounders through a balancing term in the loss. Is it possible to address the problem of confounders within this framework?

---

> ### Author Response · Authors · 2023-11-19
> **Response to 6Zs2**
>
> Thank you very much for your positive feedback on our paper! We are happy to answer your questions in the following.
>
> **Response to 6Zs2:**
>
> **# Response to Weaknesses**
>
> **Response to (1) possible downsides and robustness**
>
> Thank you. We added a **new discussion** around possible downsides and the assumptions that need to be satisfied to ensure robustness (see **new Supplement A**). Therein, we especially focus on potential limitations (e.g., explainability, scalability, etc.) that are relevant for medical practice.
>
> **Action:** We added a new discussion on the applicability in medical practice (see **new Supplement A**).
>
> **Response to (2) comparison to non-neural methods**
>
> Thank you. We are more than happy to explain why non-neural methods are **not** applicable (see Table 1).
>
> * To the best of our knowledge, the **only** applicable baseline (both neural and non-neural) is TE-CDE [1]. TE-CDE builds upon the same setting (i.e., treatment effect estimation in continuous time) as our work. However, it does **not** capture aleatoric and epistemic uncertainty. Hence, we extended TE-CDE with MC dropout [5] to obtain a baseline that is applicable to our setting.
> * There are some non-neural methods that focus on a related setting [2,3,4] but still are **not** applicable. (We cited all of them in our related work section.) There are different reasons for that: they impose strong assumptions on the outcome distribution of the potential outcomes, they are not designed for multi-dimensional outcomes and static covariate data, and scalability is limited. For example, the methods in [4] can only deal with continuous treatments but **not** discrete treatments as in our work. The Gaussian process from [3] can **not** deal with static covariates because of which all patient-level information is ignored. Further, similar to [2], it scales cubically because of which practical application in medicine is prohibited.
>
> **Action:** We have carefully checked our related work section to explain that existing, non-neural baselines are not applicable. Furthermore, we reiterate the downsides of existing baselines and the strengths of our proposed method in our new discussion (see **new Supplement A**).
>
> **# Responses to Questions**
>
> **Response to (1) clarification of robustness**
>
> Thank you. Upon reading your comment, we found that we should remove “fairly” in our conclusion. Rather, we now point to our experiments in Figure 4 where we show that our method is more robust to noise in the data-generating process as compared to the baseline.
>
> **Action:** We improved our wording in the conclusion to be more precise (see the conclusion revised **Section 5**).
>
> **Response to (2) discussion of failure modes**
>
> Thanks for this helpful suggestion. We added a **new discussion** around possible failure modes (see **new Supplement A**). Therein, we especially focus on potential weaknesses (e.g., explainability, scalability, etc.) that are relevant to medical practice.
>
>
> **Action:** We added a new discussion on the applicability in medical practice (see **new Supplement A**).
>
> **Response to (3) abstract**
>
> Thank you. We fixed our statement. You are correct: we were referring to uncertainty-aware, **neural** methods for time-varying treatment effect estimation.
>
> **Action:** We clarified the abstract by adding the term “neural”.
>
> **Response to (4) confounding**
>
> Thank you for your suggestion. The key motivation of our work was to highlight the effectiveness of combining neural CDEs and neural SDEs for uncertainty-aware prediction of the potential outcomes.
>
> To address your point, we added an extension of our method where we used balanced representations to deal with confounding. Thereby, we are consistent with the baseline (i.e., TE-CDE [1]), which also uses balanced representation. Our results demonstrate that our extended method outperforms the baseline with balanced representations.
>
> **Action:** We added an extension of our method with balanced representations (see **new** **Supplement M**). We find that our proposed method outperforms the baseline with balanced representations on a new real-world and a new semi-synthetic data set.
>
> [1] Nabeel Seedat, Fergus Imrie, Alexis Bellot, Zhaozhi Qian, and Mihaela van der Schaar. Continuous-time modeling of counterfactual outcomes using neural controlled differential equations. In ICML, 2022.
>
> [2] Yanbo Xu, Yanxun Xu, and Suchi Saria. A non-parametric Bayesian approach for estimating treatment-response curves from sparse time series. In ML4H, 2016
>
> [3] Peter Schulam and Suchi Saria. Reliable decision support using counterfactual models. In NeurIPS, 2017.
>
> [4] Hossein Soleimani, Adarsh Subbaswamy, and Suchi Saria. Treatment-response models for counterfactual reasoning with continuous-time, continuous-valued interventions. In UAI, 2017.
>
> [5] Yarin Gal and Zoubin Ghahramani. Dropout as a Bayesian approximation: Representing model uncertainty in deep learning. In ICML, 2016.

---

> > ### Comment · Reviewer_6Zs2 · 2023-11-22
> > **Response to the authors**
> >
> > I thank the authors for putting in the effort and carefully addressing questions and comments. I appreciate the new sections in the appendix and believe the changes made in response to several reviews made the paper more clear and complete. I keep my recommendation of accepting the paper.

---

### Official Review · Reviewer_UGox · 2023-11-01

**Soundness:** 2 fair
**Presentation:** 2 fair
**Contribution:** 2 fair
**Rating:** 6
**Confidence:** 3

**Summary:**

The paper introduces BNCDE which combines neural controlled differential equations with neural stochastic differential equations to model the time dimension. It leverages the Bayesian paradigm to account for both model and outcome uncertainty, allowing for uncertainty-aware treatment effect estimation in continuous time.

**Strengths:**

The paper focuses on the crucial task of estimating treatment effects over time in the context of personalized medicine.

The incorporation of uncertainty quantification is a critical aspect of the proposed methodology. It allows for probabilistic estimates of treatment effects, which is essential for making informed decisions in medical contexts.

The problem setting appears to be innovative and unique.

**Weaknesses:**

1/ It appears that the timestamps follow a point process. Is there a method to verify if the intensity satisfies the Overlap assumption: 0 < λ(t | H_t^i) < 1? I understand that intensities in point processes can exceed 1. I believe that the number of events would be less frequent with the constraint λ(t | H_t^i) < 1. Could you please provide some insights on this?

2/ Understand that the variance of the variational posterior is often smaller than that of the true posterior. Could this potentially lead to overconfidence in uncertainty predictions?

3/ Experimental results are only on one synthetic data. Is there any real-life data or semi-synthetic data that is applicable to this model?

4/ Page 7: "we the maximize ELBO"

**Questions:**

Please see section Weaknesses

---

> ### Author Response · Authors · 2023-11-19
> **Response to UGox (i)**
>
> Thank you very much for your helpful feedback on our paper. We appreciate that you find our paper crucial, innovative, and unique.
>
> **Response to (1) overlap assumption & intensity process:**
>
> Thank you for your question. Upon reading your questions, we realized that we should have spelled out more explicitly how point processes are used. You are correct: the observation times of the covariates $X_t$ and outcomes $Y_t$ follow a point process with a, possibly history-dependent, intensity function, say $\zeta(t)$. However, the point process for the observation times is **different** from the point process in the identifiability assumption. The latter is a point process with intensity $\lambda(t | H_t)$, where $\lambda(t | H_t)$ is the intensity function that governs the treatment assignment. Intuitively, this can be thought of as the analog of the propensity score in the static setting. For any given time $t$, we want the probability of treatment to be greater $0$ to ensure identifiability. In the continuous-time setting, this means that the intensity function $\lambda(t|H_t)$ is non-zero. If the intensity function was hypothetically zero, then a patient with history $H_t$ would have had no opportunity to receive treatment at time $t$. Instead, if  $\lambda(t|H_t)>0$, there is a chance that treatment is administered.
>
> We can attempt to **verify** this assumption in practice empirically. In the static setting, one would simply estimate the propensity score from data (see, e.g., [1]). We can also adopt the same strategy in our continuous-time setting. Here, we can simply estimate the intensity function from data, for example, via maximum likelihood or nonparametric approaches [2]. Thereby, we can verify whether the intensity satisfies the overlap assumption.
>
> **Action:** We clarified that the observation times of $X_t$ and $Y_t$ follow a point process $\zeta(t)$ that is different from the treatment assignment process $\lambda(t)$ (see our revised **Section 3**). We further explained how the treatment intensity can be estimated from data, so that the overlap assumption can be verified (see our revised **Section 3**).
>
> **Response to (2) posterior variance**
>
> Thank you for asking this important question. In theory, the variational weight posterior variance could also be smaller than the true weight posterior variance. In such cases, uncertainty estimates will be overconfident. This has two implications: if practitioners rely on uncertainty estimates as a decision rule to administer a treatment, they may be misled. Further, the posterior predictive distribution will become overconfident.
>
> However, our empirical findings show that this is **not** an issue for our BNCDE. Across all experiments, we find that the posterior predictive distribution of our BNCDE tends to be **conservative** (see, e.g., Figure 2 and the additional studies in **Supplements K, L, M**). That is, it tends to provide **more coverage** than the predictive posterior credible intervals should guarantee from a frequentist point of view. Further, although we cannot assess the true, underlying posterior variance, we highlight that Figure 4 clearly shows the **informativeness** of our estimated weight posterior variance. That is, we observe meaningful differences in magnitude, which demonstrates the effectiveness of our method.
>
> During hyperparameter tuning of our BNCDE, we further noticed that a higher diffusion hyperparameter in the latent SDEs generally led to **more conservativeness** in the potential outcome estimation. We therefore suggest that a careful choice of the diffusion hyperparameter is one of the most important hyperparameters in our method for applications in medical scenarios.
>
> **Action:** We added the above recommendations for choosing the diffusion hyperparameter to our implementation details (see revised **Supplement I**). Furthermore, we discuss that a larger diffusion hyperparameter leads to more conservative estimates of the posterior distribution (see new **Supplement A**).

---

> > ### Author Response · Authors · 2023-11-19
> > **Response to UGox (ii)**
> >
> > **Response to (3) applicability to other data sets**
> >
> > Thank you for giving us the opportunity to demonstrate the effectiveness of our BNCDE for other datasets. In particular, we followed your suggestions and evaluated our method using both a real-world dataset and a semi-synthetic dataset from medicine. Our real-world dataset is MIMIC III [4], which includes patients from intensive care units. Further, we followed [3] and developed a semi-synthetic dataset based on MIMIC III. The results show that our method is clearly suitable for other datasets and outperforms the baseline.
> >
> > **Action:** We added **new experiments** **using real-world data and semi-synthetic data** (see **Supplement M**). Again, our experiments show that our BNCDE consistently outperforms the baseline and yields very reliable results.
> >
> > **Response to (4) typo:**
> >
> > Thank you very much for pointing this out. We fixed the typo.
> >
> >
> >
> >
> > [1] Jonas Schweisthal, Dennis Frauen, Valentyn Melnychuk, Stefan Feuerriegel. Reliable off-policy learning for dosage combinations. In NeurIPS, 2023.
> >
> > [2] Lawrence M. Leemis. Nonparametric Estimation of the Cumulative Intensity Function for a Nonhomogeneous Poisson Process. Management Science, _37_(7), 886–900, 1991.
> >
> > [3] Valentyn Melnychuk, Dennis Frauen, and Stefan Feuerriegel. Causal transformer for estimating counterfactual outcomes. In ICML, 2022.
> >
> > [4] Alistair E. W. Johnson, Tom J. Pollard, Lu Shen, Li-wei H. Lehman, Mengling Feng, Mohammad Ghassemi, Benjamin Moody, Peter Szolovits, Leo Anthony Celi, and Roger G. Mark. MIMIC-III, a freely accessible critical care database. Scientific Data, 3(1):160035, 2016.

---

### Official Review · Reviewer_CatB · 2023-11-01

**Soundness:** 3 good
**Presentation:** 3 good
**Contribution:** 3 good
**Rating:** 6
**Confidence:** 3

**Summary:**

This paper proposed to estimate the treatment effect and quantify the uncertainty over continuous time. To achieve this goal, a VAE-based Bayesian method was proposed to learn ODE (resp. SDE) over latent variables (resp. encoder parameters), as well as the posterior distribution of potential outcomes. An evidence lower bound was derived for optimization. Experiments on semi-synthetic data were conducted.

**Strengths:**

The treatment effect over continuous time is a very important and interesting topic since it can model the instantaneous effect that is beyond the scope of traditional causal modeling such as Granger causality. Besides, this paper conducted a thorough experimental analysis regarding the uncertainty quantification. The paper is well-written and easy to understand.

**Weaknesses:**

This paper fails to discuss the motivation for learning latent representations, which was commonly used for unstructured data. However, the experimental setting in this paper is still structured data. In this regard, why do not directly model the ODE over (treatment, covariates, and outcome)? Moreover, the introduction of latent variable $Z$ may violate the unconfoundness assumption. Specifically, the implicit causal assumption behind latent representation is $Z_t$ affects $(A_t, X_t, Y_t)$ and $Z_t \to Z_{t'} (t' > t)$ which also effects $(A_{t'}, X_{t'}, Y_{t'})$. In this regard, an unblocked path will open from $Y_t$ and $Y_{t'}$. Besides, since $Y_t \perp A_t | X_t$, we should implement backdoor adjustment for $X_t$, which is necessary to eliminate the confounding bias. However, I fail to see such an adjustment in the paper.

**Questions:**

Please see the weakness above.

---

> ### Author Response · Authors · 2023-11-19
> **Response to CatB (i)**
>
> Thank you very much for your constructive feedback on our paper. We appreciate that you find the topic and the analysis in our paper interesting.
>
> **Response to (1a) latent representations:**
>
> Thank you for your question. We realized that our paper was lacking motivation for the use of latent representations. Therefore, we are happy to give more insight on this topic.
>
> Latent representations are widely used in neural architectures, and, in particular, in the literature for time-varying treatment effect estimation. Within the literature on time-varying treatment effect estimation, there are many prominent examples: [1] and [2] use latent representations in recurrent neural network architectures, [3] in the transformer architecture, and [4] and [5] in their neural controlled differential equations. In all of the previous works, latent representations are **crucial** to capture the time-varying dynamics between treatments, outcomes, and covariations, which would otherwise be difficult to model without latent representations.
>
> There are several benefits from using latent representations in our task. Latent representations allow us to capture **(i) complex nonlinearities,** model **(ii) dependencies** between observables, and **(iii) reduce dimensionality** in the observed patient history. For example, in our new experiments in **Supplement M** on real-world and semi-synthetic data, patient covariates are very high dimensional. Here, dimensionality reduction is important to encode only relevant information.
>
> Hence, in our method, this means that $Z_t$ are used to complex patterns **in** and **between** observed outcome variables $Y_t$, covariates $X_t$, and administered treatments $A_t$. As such, latent representations are essential to capture complex interactions in the data-generating mechanism in our task.
>
> Neural CDEs use latent representations by design. We provide a detailed discussion on neural CDEs below and in a new **Supplement E**.
>
> **Action:** We revised our paper to clearly spell out the motivation for using latent representations.

---

> ### Author Response · Authors · 2023-11-19
> **Response to CatB (ii)**
>
> **Response to (1b) ODE over treatment, covariates, outcome:**
>
> Thank you for this question. Upon reading your question, we realize that we should have put more emphasis on the distinction between neural ODEs and neural CDEs and why the neural ODEs are ill-suited for our task. We are happy to provide clarification on this issue.
>
> _Why did we opt for neural CDEs instead of neural ODEs?_ Neural ODEs can be thought of as residual neural networks with infinitely many, infinitesimal small hidden layer transformations. Thus, neural ODEs are designed to describe how a system evolves, given its initial conditions (e.g., a patient’s initial health condition, an initial treatment decision). However, it is **impossible** to adjust these dynamics over time, once the neural ODE is learned. This is different from neural CDEs, which are designed to adjust for sequentially incoming information [5,6].
>
> _Why is a single neural ODE not suitable for our task?_ There are clear theoretical reasons why using a single neural ODE to model the dynamics of $(Y_t, X_t, A_t)$ is **not** suitable. If we used a neural ODE to directly model the dynamics of $(Y_t, X_t, A_t)$, this would mean that we learn an ODE that describes the evolution of $(Y_t, X_t, A_t)$, given the initial conditions $(Y_0, X_0, A_0)$. In particular, we would then make the following assumptions that would conflict with our task:
>
>
>
> 1. The health conditions of a patient are completely captured in her **initial state** $(Y_0, X_0, A_0)$. The initial state is the only variable that influences the evolution of a (neural) ODE. However, we want the patient encoding to be updated over time.
> 2. Given these initial conditions, the outcome and covariate evolution are **deterministic**. It is not possible to “update” an ODE at a future point in time to account for randomness in future observations. However, we want to account for random future events when they are measured.
> 3. The treatment assignment plan is **deterministic** and cannot be changed. That means, we would assume that sequences of treatments evolve like a deterministic process. This makes modeling of arbitrary future sequences of treatments impossible. In particular, using a neural ODE for the decoder of our BNCDE would mean that we can only try to predict the outcome in the future for a **single, static** treatment, captured in the initial state of the ODE. However, we want to model **arbitrary future sequences** of treatments.
> 4. Future observations **cannot change the dynamics** of the system and, importantly, **cannot update our beliefs** on the outcomes. However, we aim to **update our beliefs** on the outcome when patient information is recorded at later points in time.
>
> _How can our architecture based on neural CDEs address the above issues?_ Our architecture solves all of the above issues. The reason is that neural CDEs with latent representations $Z_t$ have much larger flexibility that is beneficial for our task:
>
>
> 1. Neural CDEs allow for patient **personalization at any point in time $t$**, and do not assume that all information is captured at time $t=0$.
> 2. Future observations $(Y_t,X_t,A_t)$ can **update** the neural CDE. This is crucial, because not all patient trajectories follow the same deterministic evolution.
> 3. Using a neural CDE in our decoder, we can model **arbitrary future sequences of multiple treatments**.
> 4. Incoming observations after time $t=0$ can **update our beliefs**. This makes the neural CDEs particularly suited for the Bayesian paradigm.
>
> In summary, using neural ODEs to directly model the dynamics of $(Y_t, X_t, A_t)$ has significantly lower modeling capacities, limits patient personalization, and imposes prohibitive assumptions on the real world. To this end, we argue that it is difficult – if not impossible – to adequately model the input through a joint neural ODE in our task, and, as a remedy, we opted for an approach based on neural CDE instead.
>
> **Action:** We add a **new discussion** in **Supplement E** on why neural ODEs are ill-suited for the time-varying treatment regime.

---

> > ### Author Response · Authors · 2023-11-19
> > **Response to CatB (iii)**
> >
> > **Response to (2) violation unconfoundedness assumption:**
> >
> > Thank you for bringing up this point. We understand that you assume that $Z_t$ affects $(Y_t,X_t,A_t)$, which would, indeed, violate unconfoundedness.
> >
> > This, however, is not the case. In a neural CDE, the data stream $(Y_t,X_t,A_t)$ is interpolated online in time [7], as we highlighted in **Supplement D**. This is analogous to previous works in the treatment effect literature [4]. While the continuous data stream $(Y_t,X_t,A_t)$ changes the evolution $Z_{t’}$, this is **not the case in the other direction**. $(Y_t,X_t,A_t)$ are modeled **independently** of $Z_t$. Therefore, there is **no violation** of unconfoundedness.
> >
> > **Action:** We emphasize in the introduction of our model architecture that the evolution of $Z_t$ does not affect how we model $(Y_t,X_t,A_t)$, and therefore does not violate unconfoundedness. We also added the above to our **new discussion** in **Supplement E** where we provide additional motivation for our architecture design.
> >
> > **Response to (3) necessary adjustments:**
> >
> > Thank you for your suggestion. We emphasize that the theory for proper adjustments in continuous time is still very underdeveloped. In fact, there **does not exist a single method** that is capable of properly adjusting in continuous time. Of note, inverse probability weighting [7] and G-computation [2] exhibit extremely large variance even in the discrete-time setting, making them **impracticable** for many medical scenarios. Further, these adjustments are **not applicable** in continuous time.
> >
> > However, to address your point, we added an extension of our method where we used balanced representations to address confounding. Thereby, we are consistent with the baseline (i.e., TE-CDE [4]), which also uses balanced representation. Moreover, balancing has shown empirical success in discrete time [1,3] as well. Our results demonstrate that our extended method outperforms the baseline with balanced representations.
> >
> > **Action:** We added an extension of our method with balanced representations (see **new Supplement M**). We find that our proposed method outperforms the baseline with balanced representations. We also added a careful discussion of the implications of using balanced representation for medical practice (see our **new Supplement A**).
> >
> > [1] Ioana Bica, Ahmed M. Alaa, James Jordon, and Mihaela van der Schaar. Estimating counterfactual treatment outcomes over time through adversarially balanced representations. In ICLR, 2020.
> >
> > [2] Rui Li, Stephanie Hu, Mingyu Lu, Yuria Utsumi, Prithwish Chakraborty, Daby M. Sow, Piyush Madan, Jun Li, Mohamed Ghalwash, Zach Shahn, and Li-wei Lehman. G-Net: A recurrent network approach to G-computation for counterfactual prediction under a dynamic treatment regime. In ML4H, 2021.
> >
> > [3] Valentyn Melnychuk, Dennis Frauen, and Stefan Feuerriegel. Causal transformer for estimating counterfactual outcomes. In ICML, 2022.
> >
> > [4] Nabeel Seedat, Fergus Imrie, Alexis Bellot, Zhaozhi Qian, and Mihaela van der Schaar. Continuous-time modeling of counterfactual outcomes using neural controlled differential equations. In ICML, 2022.
> >
> > [5] Patrick Kidger, James Morrill, James Foster, and Terry Lyons. Neural controlled differential equations for irregular time series. In NeurIPS, 2020.
> >
> > [6] James Morrill, Patrick Kidger, Lingyi Yang, and Terry Lyons. Neural controlled differential equations for online prediction tasks. arXiv preprint, 2106.11028, 2021.
> >
> > [7] Bryan Lim, Ahmed M. Alaa, and Mihaela van der Schaar. Forecasting treatment responses over time using recurrent marginal structural networks. In NeurIPS, 2018.

---

### Author Response · Authors · 2023-11-19
**Response to all reviewers**

Thank you for your constructive feedback on our paper and your positive comments! We have addressed all your points. Our main improvements are:



1. We included **additional results** on semi-synthetic and real-world data (see new **Supplement M**). The results again demonstrate the effectiveness of our proposed method.
2. We added an extension of our method together with **balanced representations** (see new **Supplement M**). Our results demonstrate that our method outperforms the baseline with balanced representations.
3. We added a new analysis where we analyze our method in terms of **outcome uncertainty** (aleatoric uncertainty). We find that our method generates meaningful estimates of outcome uncertainty (see new **Supplement J**), which again confirms the effectiveness of our method.
4. We demonstrate the **scalability** of our method and provide a runtime analysis (see new **Supplement N**).
5. We added a clarification of why we use latent representations and neural CDEs instead of neural ODEs in our method (see new **Supplement E).**
6. We added an extensive discussion where we elaborate on the **applicability of our method in medical practice** (see new **Supplement A**). Therein, we navigate through both strengths and limitations.

We uploaded a revised version of our paper, where we highlight key changes colored in **red**. All the changes (labeled with **Action**) will be included in the camera-ready version of our paper. Given these improvements, we are confident that our paper will be a valuable contribution to the causal inference literature and a good fit for ICLR 2024.

---

### Meta-Review · Area_Chair_PwyS · 2023-12-06

**Metareview:**

The paper proposes Bayesian neural controlled differential equation (BNCDE) for treatment effect estimation in continuous time, incorporating uncertainty quantification through a system of neural controlled and stochastic differential equations.

pros:
+ Treatment effect estimation framework for time series in continuous time.
+ Demonstrates better uncertainty estimation compared to non-Bayesian counterparts.
+ well written and clear paper

cons:
+ lack of comparison to other neural methods, ignoring a broader literature; limited to synthetic data
+ potential violations of unconfoundedness; potential overconfidence due to variational inference

**Justification For Why Not Higher Score:**

lack of thorough empirical analysis

**Justification For Why Not Lower Score:**

important problem and sound contributions

---

### Decision · Program_Chairs · 2024-01-16

Accept (poster)